# The deubiquitylating enzyme USP15 regulates homologous recombination repair and cancer cell response to PARP inhibitors

Yihan Peng[1,2,3,4], Qingchao Liao[1,5], Wei Tan[3,4], Changmin Peng[2,3,4], Zhaohua Hu[1], Yali Chen[1], Zhuqing Li[3,4], Jing Li ⓘ [3,4], Bei Zhen[1], Wenge Zhu[3,4], Xiangpan Li[2], Yi Yao[2], Qibin Song[2], Chengsheng Liu[6], Xiangdong Qi[7], Fuchu He[1] & Huadong Pei[1,2,3,4]

Poly-(ADP-ribose) polymerase inhibitors (PARPi) selectively kill breast and ovarian cancers with defects in homologous recombination (HR) caused by BRCA1/2 mutations. There is also clinical evidence for the utility of PARPi in breast and ovarian cancers without BRCA mutations, but the underlying mechanism is not clear. Here, we report that the deubiquitylating enzyme USP15 affects cancer cell response to PARPi by regulating HR. Mechanistically, USP15 is recruited to DNA double-strand breaks (DSBs) by MDC1, which requires the FHA domain of MDC1 and phosphorylated Ser678 of USP15. Subsequently, USP15 deubiquitinates BARD1 BRCT domain, and promotes BARD1-HP1γ interaction, resulting in BRCA1/BARD1 retention at DSBs. USP15 knockout mice exhibit genomic instability in vivo. Furthermore, cancer-associated USP15 mutations, with decreased USP15-BARD1 interaction, increases PARP inhibitor sensitivity in cancer cells. Thus, our results identify a novel regulator of HR, which is a potential biomarker for therapeutic treatment using PARP inhibitors in cancers.

[1] State Key Laboratory of Proteomics, National Center for Protein Sciences (Beijing), Beijing Proteome Research Center, Beijing Institute of Lifeomics, Beijing 102206, China. [2] Cancer Center, Renmin Hospital of Wuhan University, Wuhan 430060, China. [3] Department of Biochemistry and Molecular Medicine, The George Washington University School of Medicine and Health Science, 2300 Eye Street, N.W., Washington, DC 20037, USA. [4] GW Cancer Center, George Washington University School of Medicine and Health Sciences, Washington, DC 20052, USA. [5] Key Laboratory of Industrial Fermentation Microbiology, Ministry of Education, Tianjin Industrial Microbiology Key Lab, College of Biotechnology, Tianjin University of Science and Technology, No. 29, 13ST. TEDA, Tianjin 300457, China. [6] Aesthetic Surgery, Jingmei Cosmetic Surgery Clinic, Beijing 100124, China. [7] Department of Plastic Surgery, General Hospital of Southern Theater Command, PLA, Liuhua Road 111, Guangzhou 510010, China. Correspondence and requests for materials should be addressed to X.Q. (email: create_beauty@hotmail.com) or to F.H. (email: hefc@nic.bmi.ac.cn) or to H.P. (email: peihuadong@hotmail.com)

In mammalian cells, there are two prominent repair pathways that repair double strand breaks (DSBs): homologous recombination (HR) repair and non-homologous end-joining (NHEJ) mechanisms[1,2]. NHEJ is referred to as "non-homologous" because the break ends are directly ligated without homologous templates. So, NHEJ is commonly associated with the presence of insertions and deletions at DSBs[3]. HR is different from NHEJ, which needs an intact homologous template, and primarily functions in the S/G2 phases[4]. A key step in HR repair is DNA end resection, which is initiated by the MRN complex with CtIP to generate a 3′ single-stranded DNA (ssDNA) tail[5–9]. Then, the 3′ ssDNA tail is extended by Exo1 and Dna2 nucleases[10–13], which are quickly bound by replication protein A (RPA). RPA is then replaced by the DNA recombinase Rad51, which forms extended helical filaments on the ssDNA[14–17]. The resulting nucleoprotein filament is responsible for pairing the ssDNA with homologous double-stranded DNA, which serves as the template to guide DSB repair[18,19].

Breast cancer-associated gene 1 (BRCA1) is one of pivotal protein during HR[20]. BRCA1 forms at least three distinct complexes (BRCA1-A, BRCA1-B, and BRCA1-C) in cells through the association of different adaptor proteins (ABRAXAS, BACH1, and CtIP) with its C-terminal BRCT domain[21–27]. The BRCA1-A complex consists of BRCA1 in association with the ubiquitin-interacting motif containing protein RAP80, the deubiquitinylating (DUB) enzymes BRCC36 and BRCC45, MERIT 40, and ABRAXAS[21–23,25,28–31]. The BRCA1-A complex is targeted to DSBs through interaction of RAP80 with K63 poly-ubiquitin chains on H2A and H2AX[21,22,28–31]. These Lys63-linked poly-ubiquitin chains were catalyzed by RNF8 and RNF168, which are targeted by the upstream mediator MDC1[21,22,28–31]. BRCA1-B and BRCA1-C complexes promote HR through helicase activity and DNA end resection, respectively[32,33], but BRCA1-A complex is not to execute HR rather to suppress excess DNA end resection[23,32,34,35].

Besides the BRCT domain, BRCA1 function is tightly linked to its N-terminal RING domain, which binds BARD1 to form a heterodimer in cells[36]. BRCA1/BARD1 complex is required for DNA end resection during HR[17–19]. BARD1 BRCT domain binds poly (ADP-ribose) (PAR) to regulate BARD1-BRCA1 accumulation at DSBs within 20 s following laser microirradiation[37]. On the other hand, the PxVxL motif in the BRCT domain of BARD1 interacts with the chromoshadow domain of HP1γ, which binds specifically to Lys9-dimethylated histone H3 (H3K9me2)[32,38,39]. BARD1–HP1γ interaction affects BRCA1/BARD1 retention at DSBs.

BRCA1 is one of the best-known genes linked to breast cancer risk. Mutations in the BRCA1 gene were found in around 50% of familial breast cancer cases[40]. The major BRCA1 binding partner, BARD1, is also implicated in the prognosis of breast cancer[41]. Depletion of BARD1 renders DNA damage sensitivity, HR deficiency, and genome destabilization. The ablation of BARD1 in mice leads to cancer susceptibility, and probable disease-causing mutations are found in patients with breast cancer[42,43].

Because individual tumors often have unique defects in the DNA damage response (DDR) pathway, insights into the basic mechanisms by which cells repair different DNA lesions could also guide individual therapy. A successful example is the use of poly-(ADP-ribose) polymerase (PARP) inhibitors in cancer patients with BRCA1 mutations[44]. Although PARP inhibitors offer a promising strategy for individual therapy, many questions apart from clinical efficacy still remain unanswered. For example, there is compelling evidence for the utility of PARP inhibitors in ovarian cancers in the absence of BRCA mutations (germline or somatic), presumably resulting from other molecular deficiencies in DNA repair. So there is a continual demand to identify BRCA-like and other genomic signatures that may expand benefits from PARP inhibitor[45].

Deubiquitinases (DUBs) play critical roles in ubiquitin-directed signaling by catalytically removing the ubiquitin from substrate proteins. In this study, we found that the deubiquitinase USP15 plays an important role in HR and cancer cells' response to PARP inhibitors. USP15 is a member of the largest subfamily of cysteine protease DUBs, which contains two conservative cysteine (Cys) and histidine (His) boxes[46]. USP15 is previously implicated in tumor growth factor-β and one morphogenetic protein response, antiviral immune response, T cell response, and neuroinflammation[47–50]. USP15 is aberrantly regulated or mutated in many human cancers[47–51]. For example, USP15 is deeply deleted in 26.7% pancreas cancer patients, and USP15 Ser678 is mutated in endometrial cancer patients[51]. However, the detailed function of USP15 in cancer remains unclear. We found that USP15 plays an important role in HR via targeting BARD1.

## Results

**USP15 regulates HR.** We and other groups previously identified USP4 and USP11 as important components of HR[52–54]. In light of the phylogenetic connections among USP4, USP15, and USP11[55], we are interested in exploring possible DDR roles of USP15. We found that knockout (KO) of USP15 by CRISPR sensitized cancer cells to DNA-damaging agents, including camptothecin (CPT), mitomycin C (MMC), hydroxyurea (HU), and ionizing radiation (IR) (Supplementary Fig. 1a). To rule out off-target effects of the USP15 guide RNA (gRNA), we ectopically expressed wild-type (WT) USP15 in USP15-depleted cells. WT USP15 rescued the DNA damage sensitivity conferred by USP15 deficiency (Supplementary Fig. 1a). To further examine whether USP15 plays a role in DSB repair, we examined γH2AX foci formation in USP15-KO cells exposed to IR. As shown in Fig. 1a and Supplementary Fig. 1b, depletion of USP15 resulted in elevated levels of spontaneous γH2AX foci formation. Moreover, in contrast to control cells, at 24 h after IR, suppression of USP15 resulted in sustained γH2AX foci, suggesting that USP15 contributes to DNA damage repair.

Next, we examined how USP15 promotes DNA repair using integrated reporter assays for HR and NHEJ[56]. We observed a significantly compromised HR in USP15-KO cells (Fig. 1b). Conversely, we observed a minor increase in NHEJ efficiency (Fig. 1c). USP15 KO also resulted in hypersensitivity to PARP inhibitor (AZD2281) (Fig. 1d), suggesting an important role of USP15 in the HR pathway. Importantly, knockdown of USP15 had no significant effect on cell-cycle distribution in cells without DNA damage treatment, indicating that the effect of USP15 depletion on HR was not caused by cell-cycle change (Supplementary Fig. 1c).

**USP15 functions in DNA-end resection.** To examine the detailed role of USP15 in HR, we examined the accumulation of several DDR factors at DNA lesions induced by ultraviolet (UV) laser micro-irradiation in USP15-KO cells. USP15 deficiency resulted in compromised accumulation of BARD1, BRCA1, RPA, and RAD51 (Fig. 1e–h). In contrast, USP15 did not affect the recruitment of upstream regulators of BARD1/BRCA1, such as H2AX, MDC1, RNF8, and RAP80 (Supplementary Fig. 1d–g). USP15 also had no effect on 53BP1 recruitment to DSBs (Supplementary Fig. 1h). USP15 mainly affected BARD1/BRCA1 retention at DSBs at late time point (Fig. 1i). Previous studies reported that loss of 53BP1 in BRCA1-deficient cells reversed their sensitivity to PARP inhibitors, through restoration of HR efficiency[57,58]. We also found that knockdown of 53BP1 in USP15-depleted cells rescued the HR efficiency, as well as PARP

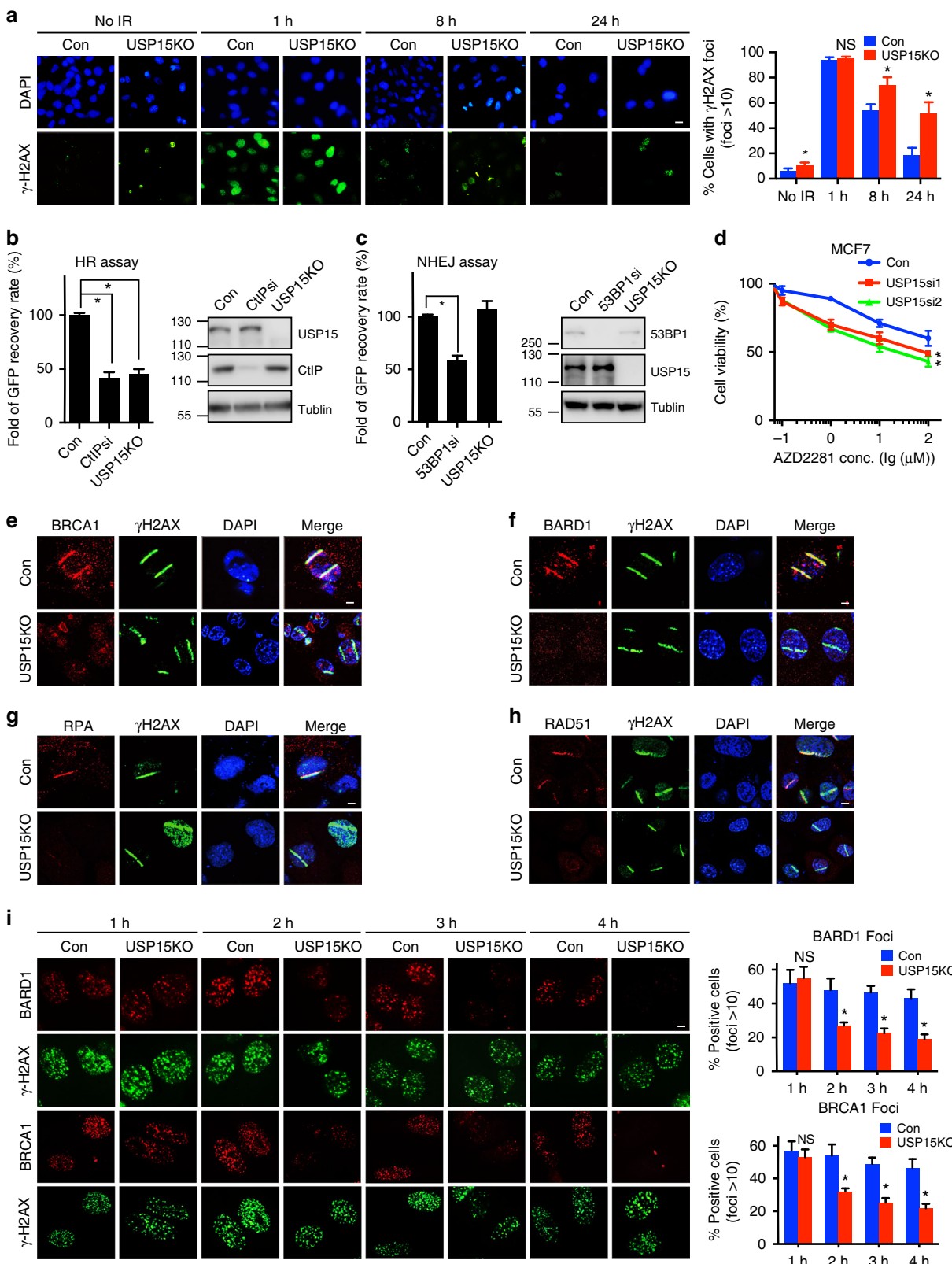

inhibitor sensitivity (Supplementary Fig. 1i, j). Based on the above results, we hypothesized that USP15 regulates DNA end resection through its effect on BARD1/BRCA1.

**USP15 interacts with BARD1 via its C-terminal region.** As USP15 affects BARD1 recruitment to DSBs, we next examined

whether USP15 could interact with BARD1. We performed reciprocal co-immunoprecipitation experiments with antibody against USP15 or BARD1. As shown in Fig. 2a, endogenous USP15 and BARD1 associated with each other in cells. Furthermore, the interaction between USP15 and BARD1 was confirmed by in vitro glutathione S-transferase (GST) pull-down assay

**Fig. 1** USP15 regulates homologous recombination (HR) and breast cancer response to poly-(ADP-ribose) polymerase (PARP) inhibitor. **a** USP15 depletion inhibits DNA damage repair. Representative γ-H2AX foci at indicated times after irradiation (2Gy) are presented (left panel). Quantification (right panel) are average of three independent experiments (100 cells each experiment), presented as mean±SD, two-tailed Student's *t* test, *$P < 0.05$. Scale bar: 40 μm. **b** HR efficiency in USP15-depleted cells were determined using the direct repeat green fluorescent protein (DR-GFP) reporter assay, as described in the Methods section. **c** Non-homologous end-joining (NHEJ) efficiency in USP15-depleted cells were determined using the EJ5-GFP reporter assay, as described in the Methods section. CtIP or 53BP1 small interfering RNAs (siRNAs) were used as a positive control, respectively, in **b**, **c**. Data are shown as average of three independent experiments, presented as mean ± SD, two-tailed Student's *t* test, *$P < 0.01$. **d** USP15 depletion sensitizes cancer cells to PARP inhibitor. The indicated MCF7 cells' response to PARP inhibitor (AZD2281) was measured by MTS assay. Data are presented as mean± EM of three independent experiments, two-way analysis of variance (ANOVA), *$P < 0.05$. **e–h** Wild-type (Con) or USP15-knockout U2OS cells were subjected to micro-irradiation as described in the Method section; 3 h later, cells were fixed and processed for immunostaining with indicated antibodies. Representative images of BRCA1 (**e**), BARD1 (**f**), RPA (**g**), and RAD51 (**h**) accumulation at sites of laser-induced DNA damage are shown. Scale bar: 10 μm. **i** Representative images of BARD1/BRCA1 foci at indicated times after irradiation (4Gy) are shown (left panel). Quantification (right panel) are average of three independent experiments, and 100 cells were counted for each experiment. Data are presented as mean±SD, two-tailed Student's *t* test, *$P < 0.01$. Scale bar: 10 μm. Unprocessed scans of blots are provided in Supplementary Fig. 8

(Supplementary Fig. 2a, b). Interestingly, USP15–BARD1 interaction was increase after IR, HU, MMC, or CPT treatment (Fig. 2b). USP15 did not associate with BRCA1 (Supplementary Fig. 2h). USP15 contains DUSP domain at its amino terminus, two UBL domains in the middle, and carboxyl terminus[55]. We next tested which region of USP15 is responsible for interacting with BARD1 by expressing BARD1 together with USP15 or its truncated mutants in HEK293T cells (Fig. 2c). USP15 deletion mutant (deletion residues 740–981) abolished the binding of USP15 with BARD1. Similarly, we generated deletion mutants of BARD1 (Fig. 2d). The USP15-binding region of BARD1 was mapped to the C-terminal BRCT domain (residues 568–777). Importantly, USP15–BARD1 interaction is essential for HR and cancer cell response to PARP inhibitor (Fig. 2e, f and Supplementary Fig. 2c–g).

**USP15 deubiquitinates BARD1 BRCT domain**. Since USP15 is a deubiquitylating enzyme, we asked whether its deubiquitylating enzyme activity is required for HR. We reconstituted USP15-KO cells with WT USP15 or a catalytically inactive mutant of USP15 (USP15-C269A). As shown in Supplementary Fig. 3a, WT USP15, but not the C269A mutant, restored HR (Supplementary Fig. 3a) and PARP1 inhibitor response (Supplementary Fig. 3b), as well as BRCA1, BARD1, RPA, and RAD51 foci formation (Supplementary Fig. 3c–g) in USP15-depleted cells. More interestingly, USP15 deubiquitinated itself, just as USP4[52,53] (Supplementary Fig. 3h, i). The catalytically inactive mutant of USP15 showed weaker binding with BARD1 (Supplementary Fig. 3h–j). These results indicate that the deubiquitinating enzyme activity of USP15 is required for its function in HR.

Since USP15 interacts with BARD1 and this interaction is important for HR, we hypothesized that BARD1 is the prime target of USP15. BARD1 ubiquitination slightly decreased after DNA damage, whereas in USP15-depleted cells, basal BARD1 ubiquitination level did not decrease (Fig. 3a). Because the C terminal of BARD1 is important for its functions in HR, we then investigated if USP15 deubiquitinates BARD1 BRCT domain in vitro and in vivo. We purified GST-USP15 WT or GST-USP15 C269A mutant from *Escherichia coli*, and did the in vitro deubiquitination assay. As shown in Supplementary Fig. 3h and Fig. 3b, USP15 deubiquitinated the BRCT domain of BARD1 in vitro. USP15 removed K63, but not K48-linked ubiquitin chains from the BRCT domain of BARD1. Interestingly, BARD1 BRCT domain ubiquitination decreased significantly following DNA damage (Fig. 3c). In addition, in cells lack of USP15, BARD1 BRCT domain ubiquitination did not decrease again (Fig. 3a, c). This effect is dependent on the deubiquitinating enzyme activity of USP15 (Fig. 3b, d). The above results indicate

that USP15 regulates HR through targeting BARD1 at the BRCT domain.

Since USP15 regulates BARD1/BRCA1 retention at DSBs (Fig. 1i), we want to elucidate the underlying mechanism. Previous studies showed that histone H2B lysine 120 mono-ubiquitination (H2BK120ub) is important for BRCA1/BARD1 recruitment to DSBs[59,60]. However, H2BK120ub did not change in USP15-KO cells (Supplementary Fig. 4a). We also found that BRCA1–BARD1 interaction and BRCA1–RAP80 interaction did not change in USP15-KO cells (Supplementary Fig. 4b). This result confirmed that USP15 did not function through RAP80. The interaction between BARD1 and poly-PAR was also unaffected in USP15-depleted cells (Supplementary Fig. 4c).

Previous studies also showed that BARD1--HP1γ interaction is required for BRCA1/BARD1 retention at DSBs[38,39]. To test whether USP15 directly promotes BARD1–HP1γ interaction through deubiquitinating the BARD1 BRCT domain, we examine BARD1–HP1γ interaction in USP15-KO cells. As shown in Fig. 3e, f, knockout of USP15 decreased BARD1–HP1γ interaction in cells, and this effect is dependent on USP15 deubiquitinating enzyme activity. GST pull-down results also showed that BARD1–HP1γ interaction is affected by the ubiquitination level of BARD1 BRCT domain (Supplementary Fig. 4d), which is regulated by USP15 (Supplementary Fig. 4e). More importantly, we made the deubiquitinated BARD1-BRCT domain by USP15, and examined its interaction with HP1γ protein in vitro. As shown in Fig. 3g, USP15 deubiquitinated BARD1 BRCT domain in vitro, and promoted BARD1–HP1γ interaction. These data strongly support that BARD1 retention at DNA damage sites is regulated by USP15 via deubiquitinating BARD1 at the BRCT domain. To further confirm this point, we knocked down BARD1 or HP1γ in USP15-KO cells, and then checked the HR efficiency and PARP inhibitor sensitivity. As shown in Supplementary Fig. 4f–i, depletion of USP15 along with BARD1 or HP1γ showed similar HR efficiency/PARP inhibitor sensitivity as compared to single depletion cells. These results clearly indicate that USP15 and BARD1/HP1γ function in the same pathway.

**USP15 is phosphorylated at Ser678 and recruited to DSBs**. We next investigated whether and how USP15 itself is regulated. As shown in Fig. 4a, USP15 was phosphorylated at ataxia telangiectasia-mutated (ATM) consensus Ser-Gln/Thr-Gln (SQ/TQ) sites after IR. No phospho-SQ/TQ signal was detected in *Atm*-deficient mouse embryonic fibroblasts (MEFs) (Fig. 4b), suggesting that USP15 is phosphorylated in an ATM-dependent manner. Previous large-scale proteomic studies also demonstrated that Ser678 of USP15 is phosphorylated by ATM following DNA damage[61]. As shown in Fig. 4c, mutation at Ser678 abolished ATM-dependent USP15 phosphorylation, indicating

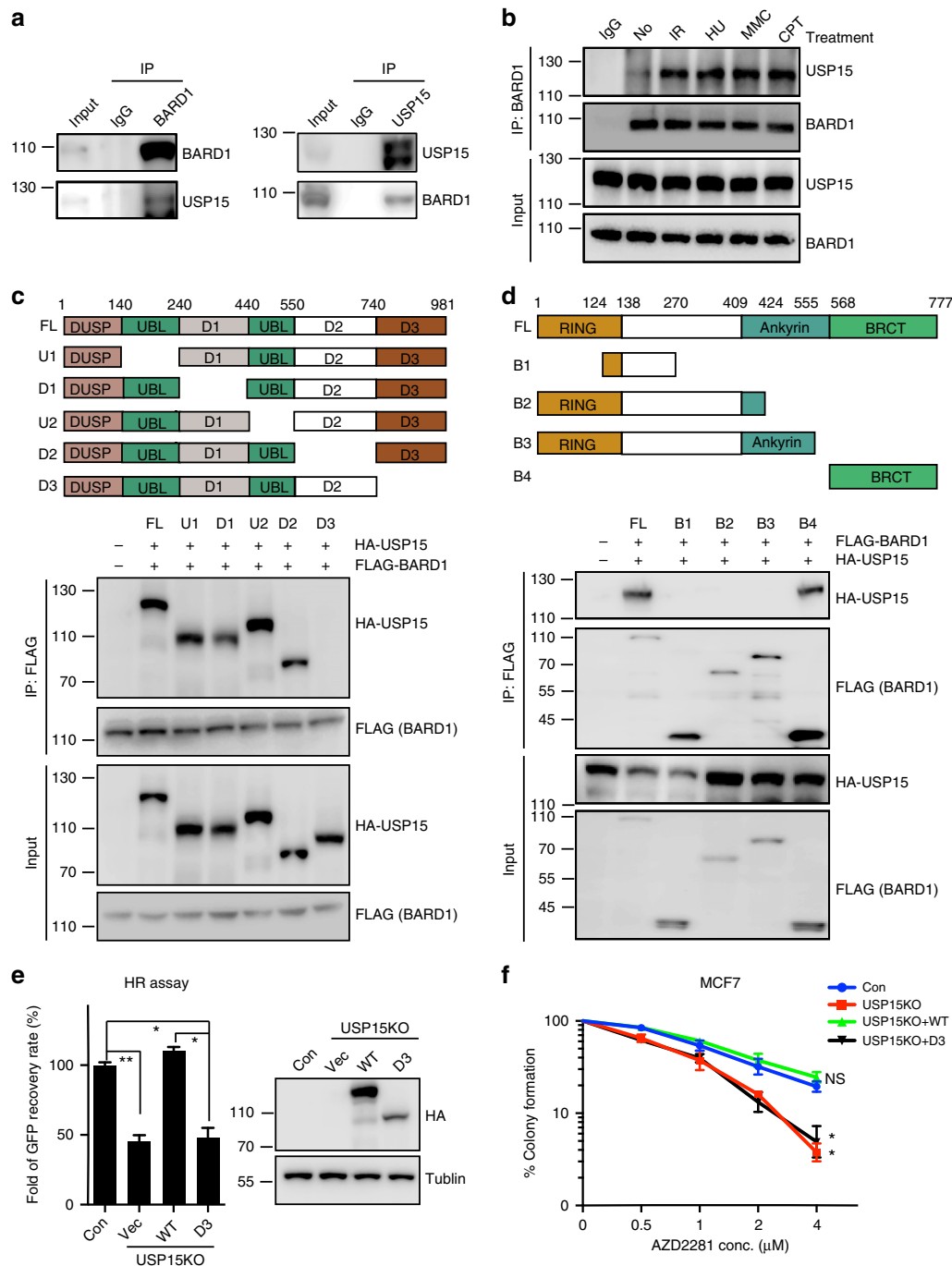

**Fig. 2** USP15 interacts with BARD1 BRCT domain through its C-terminal region. **a** Reciprocal endogenous immunoprecipitation (IP) between USP15 and BARD1 were performed in HEK293T cells. (Left panel) IP with anti-BARD1 antibody and blot with anti-BARD1 or USP15 antibody, respectively. (Right panel) IP with anti-USP15 antibody and blot with anti-USP15 or BARD1 antibody, respectively. **b** Co-immunoprecipitation (co-IP) assays were performed to check the interaction between USP15 and BARD1 upon DNA damage. HEK293T cells treated as indicated were lysed and immunoprecipitated with anti-BARD1 antibody, and Western blot was performed with indicated antibodies. **c** Schematic representation of USP15-truncated mutants used in this study (upper panel). Plasmids encoding HA-tagged full-length or deletion mutants of USP15 were co-transfected with plasmids encoding FLAG-tagged full-length BARD1 into 293T cells. Immunoprecipitation and immunoblotting were performed 48 h post transfection as indicated. **d** Schematic representation of BARD1-truncated mutants used in this study (upper panel). Plasmids encoding FLAG-tagged full-length or deletion mutants of BARD1 were co-transfected with plasmids encoding HA-tagged full-length USP15 into 293T cells. Immunoprecipitation and immunoblotting were performed 48 h post transfection as indicated. **e** USP15-knockout U2OS direct repeat green fluorescent protein (DR-GFP) cells were reconstituted with HA-USP15 wild-type (WT) or D3 mutant and homologous recombination (HR) efficiency were determined. Data are presented as mean±SD of three independent experiments. Two-tailed Student's t test, *$P < 0.05$, **$P < 0.01$. **f** The indicated cells" response to poly-(ADP-ribose) polymerase (PARP) inhibitor (AZD2281) were measured by colony formation assay. Data are presented as mean ± SEM of three independent experiments, two-way analysis of variance (ANOVA), *$P < 0.05$. Unprocessed scans of blots are provided in Supplementary Fig. 8

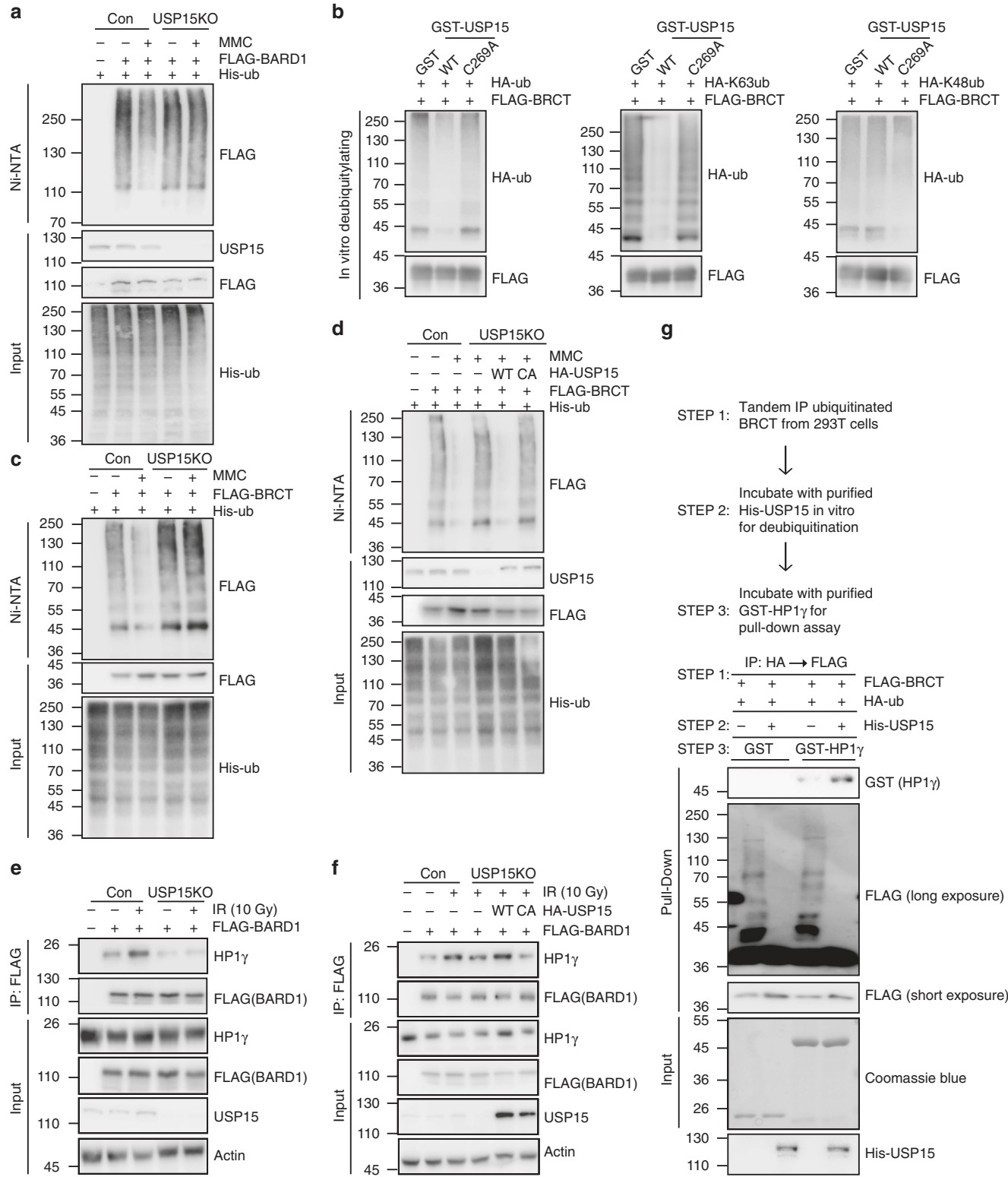

that Ser678 is the major ATM phosphorylation site of USP15. USP15 Ser678 is conserved in human, mouse and other species (Fig. 4d).

Before DNA damage, USP15 was distributed both in the nucleus and cytoplasm, with most of the signal in the cytoplasm. IR induced USP15 translocation to the nucleus (Supplementary Fig. 5a–c). We next wanted to figure out whether USP15 phosphorylation affects its localization in cells. We generated a phospho-specific antibody against Ser678, and validated the

antibodies using USP15 S678A mutant and peptide competition assays (Supplementary Fig. 5d, e). Indeed, compared to WT USP15, USP15 S678A mutant failed to translocate into nuclei following IR (Supplementary Fig. 5b, c). The phosphorylated USP15 was recruited to DSBs (Fig. 4e and Supplementary Fig. 5e), while the S678A mutant, which abolished its phosphorylation by ATM, could not be recruited to DSBs (Fig. 4e).

Mechanistically, USP15 colocalized with MDC1 upon DNA damage (Fig. 4f) and depletion of MDC1 impaired USP15

**Fig. 3** USP15 deubiquitinates BARD1 and facilitates its interaction with HP1γ. **a** Wild-type (WT) or USP15-knockout 293T cells were transiently transfected with indicated plasmids and treated with or without mitomycin C (MMC) for 24 h. Cells were lysed, and ubiquitinated BARD1 proteins were pulled down and were detected by Western blot. **b** USP15 deubiquitinates BARD1 BRCT domain in vitro. BARD1-BRCT conjugated with indicated ubiquitin chains were purified from 293T cells and were used as substrates, glutathione S-transferase (GST)-USP15 WT or GST-USP15 C269A were purified from E. coli, and then the in vitro deubiquitination assay were performed, as described in the Methods section. Samples were separated by sodium dodecyl sulfate (SDS) gel and blotted with indicated antibodies. **c** USP15 deubiquitinates BARD1 BRCT domain upon DNA damage. WT or USP15-knockout 293T cells were transfected with the indicated plasmids and treated with or without MMC for 24 h. Cells were lysed and ubiquitinated proteins were pull down by Ni-NTA. Samples were detected by Western blot with indicated antibodies. **d** USP15-knockout 293T cells reconstituted with HA-USP15 WT or C269A mutant were treated as indicated for 24 h, cells were then lysed, and ubiquitinated proteins were pull down by Ni-NTA. Samples were detected by Western blot with indicated antibodies. **e** USP15 facilitates BARD1's interaction with HP1γ. The 293T cells were transfected with the indicated plasmids and treated with or without ionizing radiation (IR) (10Gy). Cell lysates were immunoprecipitated with FLAG M2 beads and subjected to immunoblot with indicated antibodies. **f** USP15-depleted 293T cells reconstituted with HA-USP15 WT or C269A mutant were treated with or without IR (10Gy). Cell lysates were immunoprecipitated with FLAG M2 beads and subjected to immunoblot with the indicated antibodies. **g** USP15-mediated BARD1-BRCT deubiquitination promotes BARD1–HP1γ interaction in vitro. (Upper panel) Workflow of the in vitro binding assay. Briefly, ubiquitinated BARD1-BRCT were purified from 293T cells by tandem immunoprecipitation (IP) (first HA beads then FLAG beads) and were left on FLAG beads. Then one-half part was left untreated and one-half part was deubiquitinated by His-USP15 in vitro. Next, the immunoprecipitates were incubated with glutathione S-transferase (GST) or GST-HP1γ in vitro. (Lower panel) Samples were detected by Western blot with indicated antibodies. Unprocessed scans of blots are provided in Supplementary Figs 8 and 9

recruitment to DSBs (Fig. 4g). More importantly, USP15 was associated with MDC1 and this interaction was increased after DNA damage (Fig. 4h). Previous studies showed that MDC1 contains FHA domain and BRCT domain that bind phospho-SQ/TQ motif[62,63]. We also found that MDC1-FHA domain bound phosphorylated USP15, and Ser678 phosphorylation was essential for this binding (Fig. 4i, j). To further characterize the recruitment of USP15 to DSBs by MDC1, we reconstituted MDC1-deficient cells with either FHA deletion mutant or BRCT deletion mutant. WT MDC1 could restore USP15 foci formation, while both deletion mutants could not (Fig. 4k). Of note, MDC1 BRCT domain is essential for itself recruitment to DSBs[62,64], which explains why BRCT deletion mutant could not restore USP15 foci formation. Thus, ATM-dependent USP15 phosphorylation has two effects: (1) affecting the sub-cellular localization of USP15, and (2) affecting the recruitment of USP15 to DSBs.

**USP15 phosphorylation is essential for its functions in HR**. Previous studies showed that USP15 Ser678 is mutated in endometrial cancer patient[51]. To examine how USP15 phosphorylation affects cellular sensitivity to DNA damage, we reconstituted USP15-KO cells with ectopically expressed WT USP15 or USP15 S678A mutant. As shown in Fig. 5a, depletion of USP15 increased PARP inhibitor sensitivity, and reconstitution with WT USP15, but not USP15 S678A mutant, could reverse this effect. Consistent with this result, WT USP15, but not the USP15 S678A mutant, restored HR efficiency (Fig. 5b), as well as BRCA1, BARD1, RPA, and Rad51 foci formation (Fig. 5c–g). Mechanistically, USP15 S678A mutant failed to deubiquitinate BARD1 in response to MMC treatment, and in turn failed to facilitate the interaction between BARD1 and HP1γ (Supplementary Fig. 5f, g). Furthermore, ATM inhibitor (Ku55933) treatment also inhibited USP15-mediated BARD1 deubiquitinaion upon DNA damage (Supplementary Fig. 5h). These results clearly indicate that USP15 Ser678 phosphorylation is required for its function in HR.

**USP15-KO mice show genome instability**. To explore the function of USP15 in vivo, we generated USP15-KO mice in C57BL/6J strains (Supplementary Fig. 6a). We conformed that the full-length USP15 transcript was disrupted in $Usp15^{-/-}$ mice by using Western blot and reverse transcription-polymerase chain reaction (Supplementary Fig. 6b, c). The loss of USP15 full-length protein was also confirmed in $Usp15^{-/-}$ MEFs by immunoblotting (Fig. 6a) with an antibody recognizing the C-terminus region of the mouse USP15 protein. The $Usp15^{-/-}$ mice

were growth retarded. The average weight of 5-month-old $Usp15^{-/-}$ mice was 20% less as compared to that of wild-type mice (Supplementary Fig. 6d). $Usp15^{-/-}$ MEFs showed more spontaneous DNA damage as these cells displayed more γH2AX foci, which are markers of DNA damage (Fig. 6b). To test if loss of USP15 expression renders mice hypersensitive to IR, we irradiated $Usp15^{+/+}$, $Usp15^{+/-}$, and $Usp15^{-/-}$ mice. As shown in Fig. 6c, all $Usp15^{-/-}$ mice died within 17 days after 8.5Gy of irradiation, while 70% of $Usp15^{+/+}$ and 45% of $Usp15^{+/-}$ mice were still alive 1 month after irradiation. $Usp15^{-/-}$ mouse tissue showed more γH2AX signaling (Fig. 6d). On the cellular level, $Usp15^{-/-}$ MEFs exhibited defects in BARD1, RPA, and RAD51 foci formation following IR (Supplementary Fig. 6e–h), and were also hypersensitive to CPT, MMC, or AZD2281 (Fig. 6e). We also examined metaphase spreads of $Usp15^{+/+}$ and $Usp15^{-/-}$ MEFs (passage 3). Untreated $Usp15^{-/-}$ MEFs showed a nearly 3-fold increase of spontaneous single chromatid breaks compared to wild-type MEFs (Fig. 6f), suggesting an intrinsic defect in genome stability in $Usp15^{-/-}$ cells. All the above results support that USP15 functions in DSB repair in vivo.

**Cancer-associated mutations of USP15 result in HR defect**. Failure to detect and repair DNA damage leads to genomic instability, which in turn could drive tumorigenesis. Many human genetic cancer predisposition syndromes are linked to DDR protein mutations. For example, mutations in the BRCA1 gene were found in around 50% of familial breast cancer cases[40]. Previous studies also showed that USP15 Ser678 is mutated in endometrial cancer patient[51]. We confirmed that USP15 S678 phosphorylation is essential for its functions in HR and genome stability. We also investigated other potential USP15 mutations in patients with breast cancer based on the public database The Cancer Genome Atlas (TCGA) (https://portal.gdc.cancer.gov/). Interestingly, we identified two mutations of USP15 (M861V and D967H) at the C-terminal D3 region in breast cancer patients (Fig. 7a). We confirmed that this region was responsible for BARD1 binding (Fig. 2c). To investigate if these two cancer-associated mutations affect USP15 functions in genome stability. We reconstituted USP15-KO cells with ectopically expressed WT USP15 or USP15 M861V/D967H mutant. As shown in Fig. 7b, WT USP15, but not the USP15 M861V/D967H mutant, restored HR efficiency, and BARD1, RPA, and RAD51 foci formation (Fig. 7a, c–e). USP15 M861V/D967H mutant is more sensitive to PARP1 inhibitor (Fig. 7f). At the molecular level, USP15 M861V/D967H disrupted USP15-BARD1 interaction and failed to deubiquitinate BARD1 at the BRCT domain (Fig. 7g, h). Based on

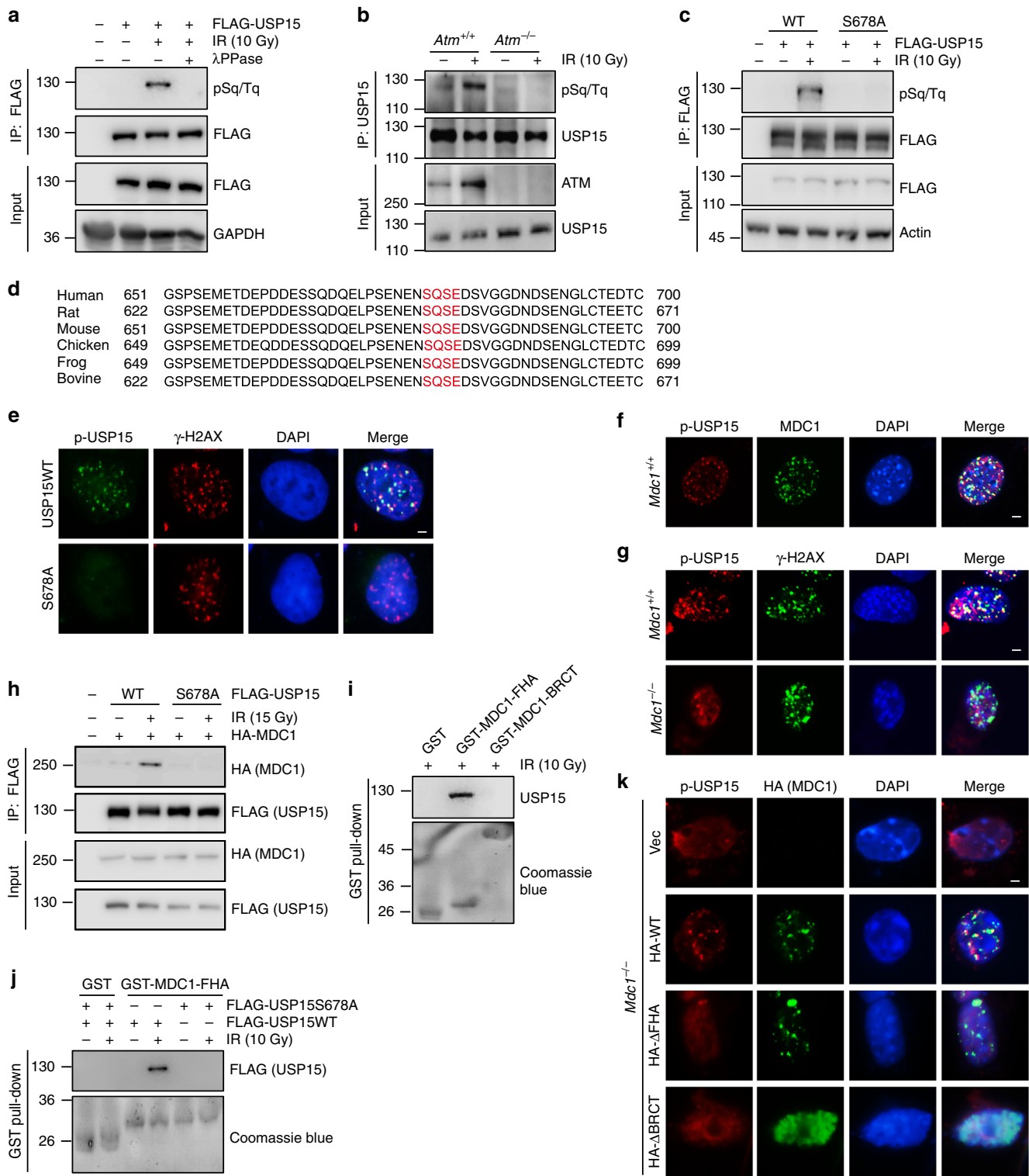

our studies, we expect that USP15 M861V/D967H contributes to breast cancer etiology, and the patients with these mutants will respond to PARP inhibitor very well. This hypothesis remains to be validated in the future.

PARP inhibitor was first approved for breast and ovarian cancer treatment, but evolving evidences suggest that PARP inhibitor can also be used to treat other cancers such as pancreatic cancer. Therefore, we next investigated whether USP15 is dysregulated in other cancer cell lines. TCGA-ACbc cohort indicated that there was deep depletion of *USP15* gene in

16.67% of breast cancer patients (Supplementary Fig. 7a). Previous studies also showed that *USP15* copy number is deeply depleted in almost 26.7% pancreatic cancer patients. Moreover, integration analysis of copy number changes in human pancreas cancer cell lines from The Cancer Cell Line Encyclopedia (CCLE: https://portals.broadinstitute.org/ccle) suggests selective down-regulation of *USP15* in pancreatic cancer cell lines (data not shown). As shown in Supplementary Fig. 7b, we also confirmed that USP15 is down-regulated in many pancreatic cancer cell lines, especially in MIAPACA2 and PK9 cell lines. Interestingly,

**Fig. 4** USP15 is phosphorylated by ataxia telangiectasia-mutated (ATM) and recruited to DNA damage sites by MDC1. **a** The 293T cells were treated and immunoprecipitated as indicated, and then analyzed with anti-pSQ/TQ antibody. **b** $Atm^{+/+}$ and $Atm^{-/-}$ mouse embryonic fibroblasts (MEFs) were treated with or without ionizing radiation (IR) (10 Gy). USP15 proteins were immunoprecipitated and blotted with anti-pSQ/TQ antibody. **c** The 293T cells transfected with the indicated plasmids were treated as indicated, and then immunoprecipitated and immunoblotted with indicated antibodies. **d** USP15 Ser678 site is conserved across species. **e** USP15-knockout U2OS cells were reconstituted with USP15 wild-type (WT) or S678A mutant. Then cells were treated with IR (4Gy), and 1 h later, stained with indicated antibodies. Scale bar: 5 μm. **f** Phosphorylated USP15 co-localizes with MDC1 upon DNA damage. $Mdc1^{+/+}$ MEF cells were treated with IR (4Gy) and allowed to recover for 1 h. Then, cells were fixed and processed for immunostaining with the indicated antibodies. Scale bar: 5 μm. **g** MDC1 depletion abrogates USP15 foci formation upon DNA damage. $Mdc1^{+/+}$ or $Mdc1^{-/-}$ MEF cells were treated with IR (4Gy) and allowed to recover for 1 h. Then, cells were fixed and processed for immunostaining with the indicated antibodies. Scale bar: 5 μm. **h** The 293T cells were transfected with indicated plasmids. The interaction between USP15 and MDC1 were investigated by immunoprecipitation and immunoblot with indicated antibodies. **i** Interactions between MDC1-FHA domain/MDC1-BRCT domain and USP15 were investigated by GST-pull down assay. **j** 293T cells were transfected with indicated plasmids and treated as indicated. The interaction between MDC1-FHA domain and USP15 was investigated by GST-pull down assay. **k** $Mdc1^{-/-}$ MEF cells transfected with indicated plasmids were irradiated, and 1 h later, stained with indicated antibodies. Scale bar: 5 μm. Unprocessed scans of blots are provided in Supplementary Fig. 9

MIAPACA2 (half-maximal inhibitory concentration (IC$_{50}$) ~3.46 μM) and PK9 (IC$_{50}$ ~4.9 μM) cells showed lower IC$_{50}$ of PARP inhibitor, compared with PANC1 cells with high USP15 (IC$_{50}$ ~52.9 μM) (Supplementary Fig. 7c), supporting that loss of USP15 expression results in hypersensitivity to PARP inhibitor. Furthermore, overexpression of USP15 in MIAPACA2 and PK9 cells rendered these cells to PARP inhibitor resistance (Supplementary Fig. 7d, e), while depletion of USP15 in PANC1 sensitize these cells to PARP inhibitor (Supplementary Fig. 7f). Consistently, overexpression of USP15 in MIAPACA2 and PK9 cells promoted BARD1/BRCA1 foci formation upon PARP inhibitor treatment (Supplementary Fig. 7g–i), while depletion of USP15 in PANC1 cells impaired BARD1/BRCA1 foci formation (Supplementary Fig. 7j). These data suggest that USP15 could serve as a biomarker for PARP inhibitor treatment in pancreatic cancers.

## Discussion

We have established that USP15 regulates HR and cancer cells' response to PARP inhibitor. Mechanistically, USP15 promotes BARD1/BRCA1 retention at DBSs that facilitates DSB end resection. Accordingly, USP15 is essential for DNA damage repair in vivo. Moreover, USP15 C-terminal is mutated in breast cancer patients, disrupting USP15–BARD1 interaction and resulting in HR defect. USP15 thus joins a growing number of proteins known to affect HR and PARP1 inhibitor response, highlighting the crucial importance of appropriately controlling and regulating the initiation of DNA end resection.

BRCA1 binds BARD1 to form a stable heterodimer in cells[65]. The retention of BRCA1/BARD1 at DSBs is important for its functions in DSBs end resection. However, the underlying mechanism is not fully understood. Previous studies reported that BARD1 C-terminal BRCT domain binds PAR within 20 s following laser microirradiation[37]. PARP inhibitors or a K619A mutation of BARD1, which abolishes the PAR interaction, disrupted the early accumulation of BRCA1/BARD1 at DSBs, bud had no effect on the BRCA1/BARD1 complex retention at DSBs at late time point. This fits with the fact that PAR signals at DSBs are transient and disappear in 10 min. These results imply that BARD1-BRCT may bind other players at late time point. Wu et al.[63] reported that the BRCT domain of BARD1 binds HP1γ. ATM, but not RNF168, regulates BARD1–HP1 interaction. But the underlying mechanism is not clear. Here we found that ATM phosphorylates USP15 after DNA damage, affects USP15 recruitment to DNA damage sites, and BARD1–HP1γ interaction, and therefore regulates BARD1/BRCA1 retention at DNA damage sites. So, PARP-mediated PAR and ATM-mediated phosphorylation regulate BARD1/BRCA1 recruitment at early time point and late time point, respectively. These two pathways coordinate together to control DNA end resection and HR.

The deubiquitinating enzymes (DUBs) play important roles in maintaining genome stability and human disease[66–68]. Among them, USP7, USP21, and USP48 have been associated with HR[66–68]. USP4, USP11, and USP15 are paralogous DUBs as evidenced by structural organization and sequence similarity. We and others previously reported that both of USP4 and USP11 function in DNA repair. Here we provided experimental evidence that USP15 is also important for genome stability maintenance in vitro and in vivo. More interestingly, the catalytically inactive USP15 was impaired in binding to BARD1 and failed to function in HR (Supplementary Fig. 3h–j). USP15 auto-deubiquitination promotes HR. Considering the similar mechanisms for USP4, perhaps this is a universal mechanism for USP family DUB function regulation.

PARP inhibitors are new class of drugs that have attracted a great deal of attention with potential in the management of patients with BRCA mutations. At the same time, there is a continual demand to search for biomarkers that can efficiently identify tumors that are most likely to respond to PARP inhibitor treatment. Perhaps, the novel gene that regulates HR also affects PARP inhibitor response. Here we found that USP15 affected HR and cancer cells response to PARP inhibitor. More interestingly, patients carried USP15 mutations that disrupted USP15-BARD1 interaction, and the cancer cell is more sensitive to PARP inhibitor. Previous other studies also showed that USP15 Ser678 is mutated in endometrial cancer patient[51]. USP15 Ser678 is phosphorylated by ATM, which is essential for USP15 recruitment to DSBs. We brought new insights for the cancer patients with USP15 mutations. Expanding and identifying populations that carried USP15 mutations affecting its functions in HR may result in improving clinical outcomes.

Furthermore, previous studies reported that *USP15* is deeply deleted in 26.7% pancreatic cancer patients. We also found that USP15 expression is lost or significantly decreased in a high percentage of the pancreatic cancer cell lines (Supplementary Fig. 7b). Loss of USP15 expression renders pancreatic cancer cells sensitivity to PARP inhibitor. Furthermore, overexpression of USP15 in a pancreatic cancer cell line (MIAPACA2) having low endogenous levels of USP15 rendered the cells resistance to PARP inhibitor. Our results implied that USP15 can also be used as a biomarker for PARP inhibitor treatment in pancreatic cancers. Overall, consistent research and progress of current studies will lead to a stronger understanding and a more comprehensive view of USP15 functions in cancer and their role in future treatment strategies.

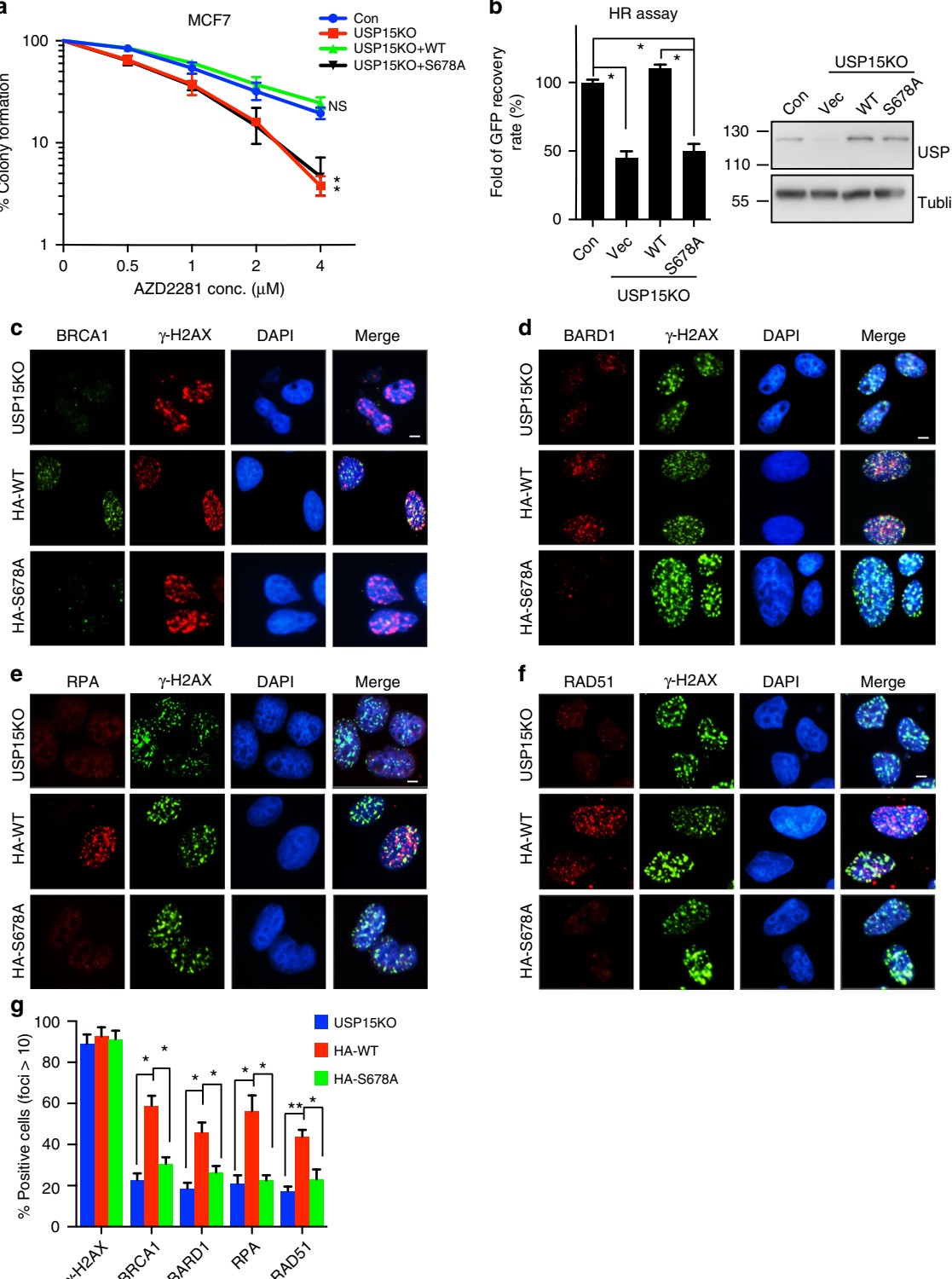

**Fig. 5** USP15 Ser678 phosphorylation is essential for its function in homologous recombination (HR). **a** USP15-depleted MCF7 cells were reconstituted with USP15 wild-type (WT) or S678A mutant. Cells' response to poly-(ADP-ribose) polymerase (PARP) inhibitor (AZD2281) were analyzed by colony formation assay. Data are presented as mean±SEM of three independent experiments, two-way analysis of variance (ANOVA), *P < 0.05. **b** USP15-knockout U2OS direct repeat green fluorescent protein (DR-GFP) cells were reconstituted with HA-USP15 WT or S678A mutant and HR efficiency were determined. Data are presented as mean±SD of three independent experiments. Two-tailed Student's *t* test, *P < 0.01. **c–g** USP15-knockout U2OS cells reconstituted with USP15 WT or S678A mutants were irradiated (4Gy), and 3 h later, BRCA1 foci (**c**), BARD1 foci (**d**), RPA foci (**e**), and RAD51 foci (**f**) are examined by indicated antibodies. Quantification (**g**) were average of three independent experiments, and 100 cells were counted for each experiment. Data are presented as mean ± SD, two-tailed Student's *t* test, *P < 0.05, **P < 0.01. Scale bar: 10 μm. Unprocessed scans of blots are provided in Supplementary Fig. 10

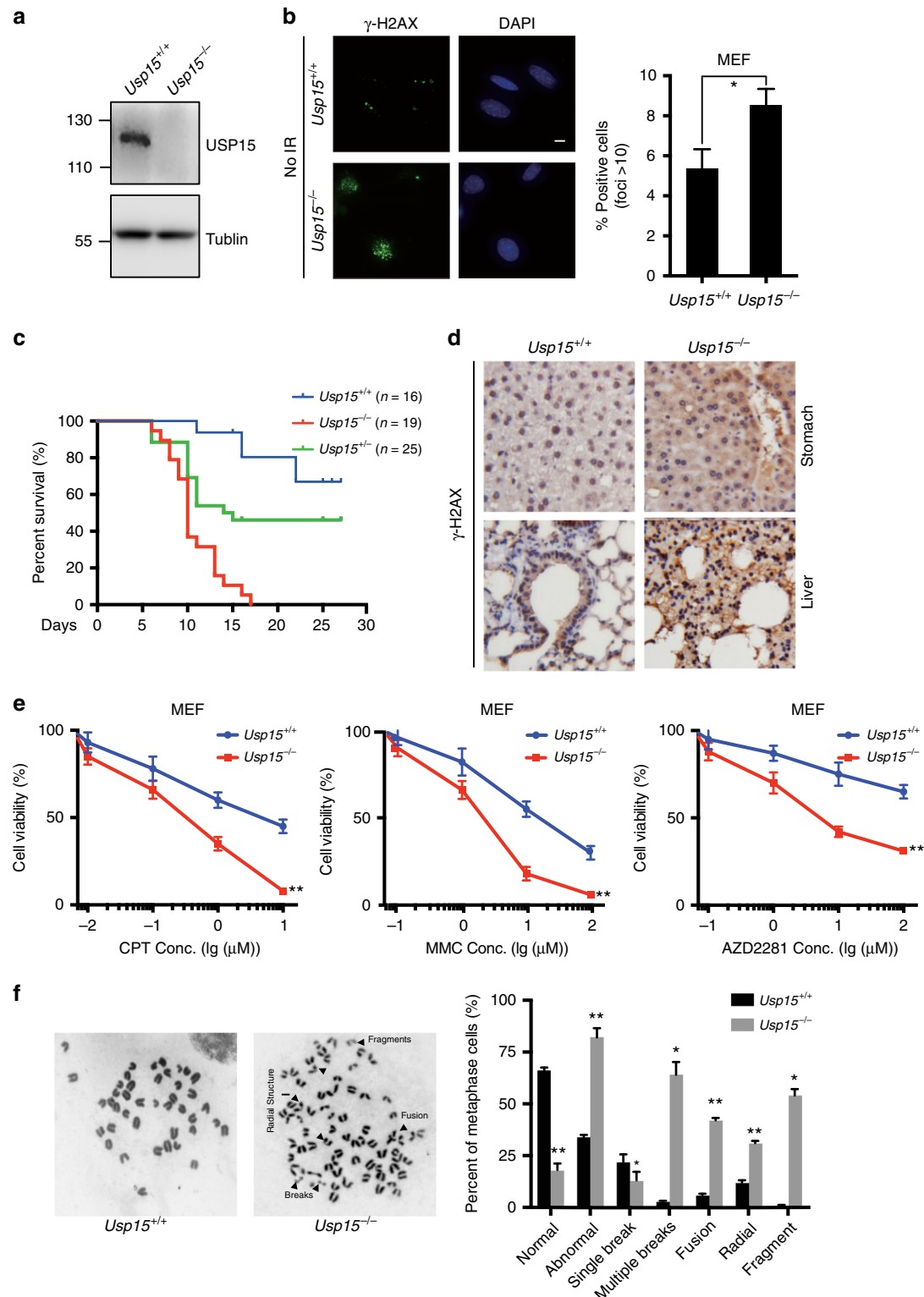

## Methods

**Cell culture**. HEK293T (ATCC, CRL-11268), U2OS (ATCC, HTB-96), and MCF7 (ATCC, HTB-22) cells were maintained in Dulbecco's modified Eagle's medium (DMEM) supplied with 10% fetal bovine serum (FBS) and penicillin/streptomycin (P/S) at 37 °C with 5% $CO_2$. $Usp15^{+/+}$ and $Usp15^{-/-}$ MEF cells were obtained from E13.5 embryos, and were maintained in DMEM supplied with 15% FBS and P/S. U2OS DR-GFP (direct repeat green fluorescent protein) and U2OS EJ5-GFP cell lines were kindly provided by Dr. Jeremy Stark[56].

USP15 KO, HEK293T, U2OS, and MCF7 cells were generated via CRISPR/ CAS9, using Lenti CRISPR V2 containing a gRNA that targeted USP15

(GCGTCGCGATGTCAGACCGC). USP15-KO cells were maintained in DMEM supplied with 10% FBS and P/S.

**Generation of *Usp15*-null mice**. All experiments with mice followed protocols approved by the Animal Care and Use Committee of the Beijing Proteome Research Center. *Usp15*-KO mice were generated from C57BL/6J strains. The scheme of *Usp15* KO was depicted in the Supplementary Fig. 6a. Briefly, *Usp15* gRNA and CAS9 mRNA were micro-injected into C57BL/6J zygote to generate germ line *Usp15*-KO mice. Mouse *Usp15* gene was frame shifted by excluding

**Fig. 6** USP15 maintains genome stability in vivo. **a** Usp15 protein is absent in $Usp15^{-/-}$ mouse embryonic fibroblasts (MEFs). Cell lysates from $Usp15^{+/+}$ and $Usp15^{-/-}$ MEFs were separated by sodium dodecyl sulfate (SDS) gel and blotted with indicated antibodies. **b** Representative γ-H2AX foci of untreated primary $Usp15^{+/+}$ or $Usp15^{-/-}$ MEF cells (passage 2) are shown (left panel). Scale bar: 20 μm. Quantification are mean±SD of three independent experiments, and 100 cells per experiment were counted. Two-tailed Student's t test, *$P < 0.05$. **c** Five-week-old $Usp15^{+/+}$ ($n = 16$), $Usp15^{+/-}$ ($n = 19$), and $Usp15^{-/-}$ ($n = 25$) littermates were subject to 8.5Gy X-ray radiation, mice were monitored daily, and survival was assessed by Kaplan–Meier curve and log-rank test. **d** Fourteen days after mice were treated as in **c**, they were dissected and tissues were subject to immunochemistry (IHC) with γ-H2AX antibody. Representative γ-H2AX IHC staining of indicated tissues are shown. **e** MTS assay of $Usp15^{-/-}$ MEFs compared with $Usp15^{+/+}$ MEFs in response to increasing doses of camptothecin (CPT), mitomycin C (MMC), and poly-(ADP-ribose) polymerase (PARP) inhibitor (AZD2281) were performed. Data are presented as mean ± SEM of three independent experiments. **f** Metaphase spreads to analyze chromosome aberrations in untreated primary $Usp15^{+/+}$ or $Usp15^{-/-}$ MEFs (Passage 3) were performed (left panel). Quantification of indicated aberrations are mean of three independent experiments, and 100 metaphases per experiment were counted. Two-tailed Student's t test, *$P < 0.05$, **$P < 0.01$. Unprocessed scans of blots are provided in Supplementary Fig. 10

Exon2. Then, the homozygous Usp15-KOknockout mice were separated from chimeric mice by crossbreeding. The following small gRNAs were used for mouse Usp15 KO:

Crispr#1:5′-CCGCTGCTTCGGCCTGCATTTGG-3′; Crispr#2:5′-CCCACTTGCCGGAACTCCTCCAG-3′.

Usp15 gene KO were identified and verified by genotyping, sequencing, and Western blot. Mouse tail genotyping was performed using mouse genotyping kit (KAPA Biosystems) and primers as follows:

Primer K forward: 5′-CCACTGTGAGGGAGGGAGACTGCT-3′
Primer K reverse: 5′-CCTGGTCCATTCAGTGCTCCACAT-3′
Primer W forward: 5′-CCACTGTGAGGGAGGGAGACTGCT-3′
Primer W reverse: 5′-ATGGGCTGATGTAAGAATGGAAAGTGAT-3′

**Plasmids and siRNAs**. USP15 full length cDNA or truncated mutants were subcloned into a modified pIRES2-S-SBP-FLAG vector or HA-tagged vector (pCMV-HA). USP15 full length cDNA were cloned into pET-28a for His tagged USP15 expression. USP15 full length cDNA or truncated mutants were subcloned into pCDH to generate Lenti-virus expression plasmids. All mutants were generated by site-directed mutagenesis and confirmed by sequencing. Full length and truncated HA-MDC1, GST-MDC1-FHA and GST-MDC1-BRCT were kind gifts from Dr. Zhenkun Lou. GST-HP1γ was a gift from Naoko Tanese (Addgene plasmid # 24076). All qRT-PCR were done by MonAmp™ SYBR Green qPCR Mix (RN04005M, Monad Biotech Co., Ltd, Wuhan, China) or MonScript™ RTIII All-in-One Mix (with dsDNase) (RN05004M, Monad Biotech Co., Ltd, Wuhan, China)

Small interfering RNA (siRNA) used in this study were as follows:

USP15si1: 5′-GATGAUACCAGGCAUAUAA-3′
USP15si2: 5′-GGUAUUGUCCGAAUUGUAA-3′
BARD1si: 5′-AGCUGAAUAUUAUACCAGA-3′
CtIPsi: 5′-CGGCAGCAGAAUCUUAAACUU-3′
HP1γsi: 5′-AGACAGCAGUGGAGAAUUG-3′
53BP1si: 5′-GAAGGACGGAGUACUAAUA-3′

**Antibodies**. Antibodies used in this study were as follows: anti-USP15 (A300-923A, WB: 1:1000), anti-BARD1 (A300-263A, WB: 1:1000; IF: 1:200), and anti-RAP80 (A300-764A, WB: 1:1000; IF: 1:100) were purchased from Bethyl Laboratory. Anti-HP1γ (MABE656, WB: 1:2000). Mouse anti-γ-H2AX (05-636, IF: 1:500), mouse anti-MDC1 (05-1572, IF: 1:300) and mouse anti-FK2 (04-263, IF: 1:200) were purchased from Millipore. Anti-RPA 70 (#2267, IF: 1:200), rabbit anti-γ-H2AX (#9718, IF: 1:400), and rabbit anti-HA (#3724, WB: 1:2500; IF: 1:300) were from Cell signaling Technology. Anti-Rad51 (GTX100469, IF: 1:200) were from Genetex. Anti-BRCA1 (D9, IF: 1:50) and anti-His tag (sc8036, WB: 1:1000) were from Santa Cruz. Anti-poly PAR (4336-BPC-100, WB: 1:1000) were from Trevigon. Anti-53BP1 (NB100-304, WB: 1:1000; IF: 1:300) were from Novus Biologicals. Anti-RNF8 (14112-1-AP, IF: 1:100) were from Proteintech. Rabbit anti-FLAG (F7425, WB: 1:2500) and mouse anti-FLAG (F3165, WB: 1:2500) were from Sigma-Aldrich. Mouse anti-HA (901501, WB: 1:2500; IF: 1:100) were from BioLegend.

**HR and NHEJ reporter assay**. U2OS cells integrated with DR-GFP or EJ5-GFP cassettes were used in the analysis of HR or NHEJ efficiency, respectively[56]. Cells transiently transfected with indicated plasmids or siRNA were then transfected with I-SceI expression vector pCBASceI (Addgene). Forty-eight hours after transfection, the percentage of GFP-positive cells were analyzed by flow cytometry. HR or NHEJ efficiency was presented as the percentage of control cells. HR or NHEJ frequencies presented in figures are the mean ± SD of three independent experiments.

**Laser micro-irradiation**. Wild-type or USP15-KO U2OS cells were seeded in glass-bottom dishes and were pre-sensitized with Hoechst33342 for 30 min and then were subject to a 405-nm localized laser beam (70% laser power, 10 s) on an inverted Nikon A1R microscope. Following laser irradiation, cells were allowed to recover for

the indicated time. Cells were then fixed with 4% paraformaldhye for 10 min at room temperature. Immunofluorescence staining was performed, and cells were then imaged using the Nikon TIE microscope and the Nikon NIS software.

**Immunofluorescence staining**. U2OS cells were seeded on poly-lysine-coated coverslips and were subject to indicated treatment. To examine BARD1, RAP80, RAD51, and phospho-USP15 foci formation, cells were pre-treated with 0.5% Triton X-100 solution on ice for 5 min prior to fixation. To detect other DDR factors foci formation, cells were fixed in 4% paraformaldehyde for 15 min and permeabilized in 0.5% Triton X-100 solution for 5 min at room temperature. Cells were then blocked with 5% bovine serum albumin solution for 1 h and incubated with primary antibody overnight at 4 °C. The coverslips were washed three times with phosphate-buffered saline (PBS), and secondary antibody was applied for 1 h at room temperature. The coverslips were then mounted onto glass slides using 4′,6-diamidino-2-phenylindole containing anti-fade solution. All the samples were visualized by a Nikon NIE fluorescence microscope and the Nikon NIS software. Unprocessed immunofluorescence images have been deposited in the figshare.

**Immunoprecipitation assay**. Cells transfected with indicated plasmids were collected and lysed in NETN buffer (10 mM Tris-HCl [pH 8.0], 100 mM NaCl, 1 mM EDTA, and 0.5% NP-40) with protease inhibitors (Roche) on ice for 30 min. Then, cell lysates were subject to FLAG M2 (Sigma Aldrich), S-protein beads (Millipore), or HA beads (Sigma Aldrich). After rotation for 8 h at 4 °C, beads were washed with NETN buffer three times, and samples were boiled with 2× sodium dodecyl sulfate (SDS) loading buffer and were subject to immunoblot with indicated antibodies. For endogenous immunoprecipitation (IP), cell lysates were incubated with indicated antibody at 4 °C for 6 h, and then were subject to Protein A/G beads (Thermo fisher) for 4 h at 4 °C. Beads were then washed with NETN buffer three times, and samples were boiled with 2× SDS loading buffer, and were analyzed by immunoblot with indicated antibodies.

**Denatured deubiquitination assay in vivo**. HEK293T cells were transfected with His-tagged ubiquitin expression plasmids along with other indicated plasmids. Cells were treated with 10 μM MG132 (Sigma-Aldrich) for 8 h and lysed in the lysis buffer (6 M guandinium-HCl, 0.1 M $Na_2HPO_4/NaH_2PO_4$, 10 mM Tris-HCl [pH 8], 5 mM imidazole, 0.01 M β-mercaptoethanol) for 30 min. The ubiquitinated proteins were pulled down by Ni-NTA (GE Healthcare) at 4 °C overnight. Then, the beads were washed twice with the lysis buffer, twice with the wash buffer (0.1 M $Na_2HPO_4/NaH_2PO_4$, 10 mM Tris-HCl [pH 6.8], 0.005 M imidazole, 0.01 M β-mercaptoethanol), and twice with the wash buffer with 0.1% Triton X-100. Samples were boiled in 2× SDS loading buffer with 0.3 M imidazole and were detected by Western blot.

**Protein purification**. For GST-tagged protein purification, GST fusion protein were expressed and purified from BL21 E. coli bacteria and were immobilized on glutathione sepharose 4B (GE Healthcare) at 4 °C overnight. Proteins were then eluted with 5-fold volume of glutathione solution (Sigma-Aldrich), and were condensed using 10 kDa protein concentrators (Thermo Fisher). Proteins were resolved and stocked in PBS containing 5% glycerol at −80 °C.

For ubiquitinated BARD1-BRCT domain purification, 293T cells were co-transfected with FLAG-BARD1-BRCT and HA-Ub. Cells were lysed in SDS buffer with protease inhibitors in denaturing condition. Ubiquitinated proteins were first purified with HA beads. Then, proteins were eluted with 5-fold 100 μg/ml HA peptide solution (Sigma-Aldrich) and ubiquitinated BARD1-BRCT proteins were further purified from the elute with FLAG M2 beads.

**GST pull-down assay**. GST fusion proteins were purified from BL21 E. coli bacteria and were immobilized on glutathione sepharose 4B at 4 °C overnight. HEK293T cells transfected with indicated constructs were treated as indicated. Then, cells were lysed in the NETN buffer with the protease inhibitor and were incubated with the

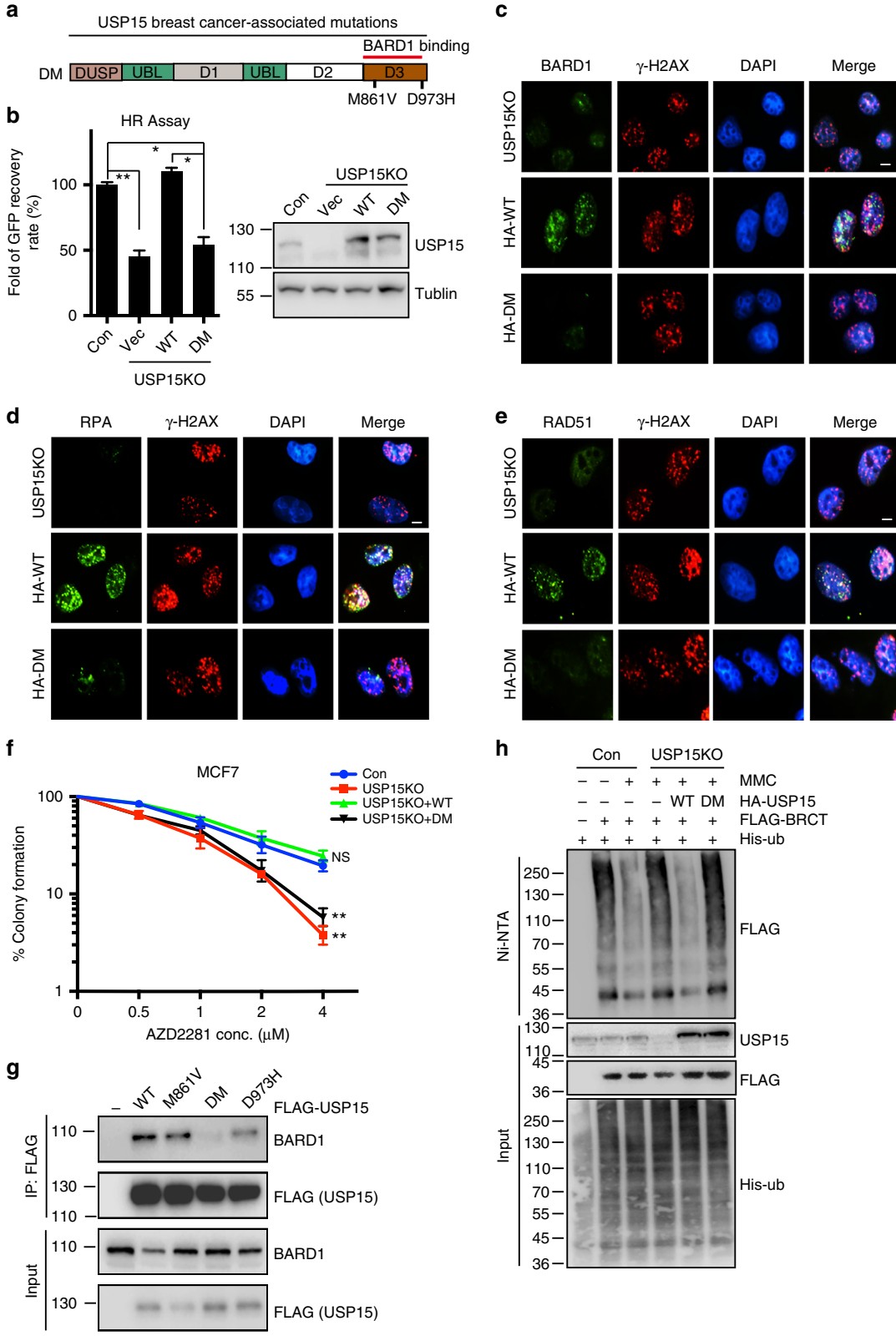

Sepharose immobilized with indicated proteins at 4 °C for 8 h. Sepharose were then washed with the NETN buffer three times and were boiled in 2× SDS loading buffer. Samples were subject to immunoblot with indicated antibodies.

**in vitro deubiquitination assay**. GST fusion USP15 or His-tagged USP15 proteins were expressed and purified from *E. coli* BL21 cells. Purified proteins were prepared in reaction buffer (50 mM Tris-HCl [pH 8.0], 50 mM NaCl, 1 mM EDTA, 10 mM

DTT, 5% glycerol). FLAG-BARD1-BRCT conjugated with HA-Ub were purified from HEK293T cells and were immobilized on FLAG M2 beads. Then, the enzyme and substrate were incubated in the reaction buffer at 37 °C for 2 h. Samples were analyzed by Western blot with indicated antibodies.

**Generation and validation of phospho-USP15 (Ser678) antibody**. Rabbit phospho-USP15 (Ser678) antibody were produced by Gen Script (Nanjing, China)

**Fig. 7** The cancer-associated USP15 mutation resulted in homologous recombination (HR) defect. **a** A schematic representation of breast cancer-associated USP15 mutations. **b** USP15-knockout U2OS direct repeat green fluorescent protein (DR-GFP) cells were reconstituted with HA-USP15 wild-type (WT) or deletion mutant (DM) and HR efficiency were determined. Data are presented as mean±SD of three independent experiments. Student's $t$ test, *$P < 0.05$, **$P < 0.01$. **c–e** USP15-knockout U2OS cells stably reconstituted with USP15 WT or DM mutants were subjected to IR (4Gy). Representative immunostaining of BARD1 (**c**), RPA (**d**), and RAD51 (**e**) are shown. Scale bar: 10 μm. **f** USP15-depleted MCF7 cells were reconstituted with USP15 WT or DM mutant. Cells' response to poly-(ADP-ribose) polymerase (PARP) inhibitor (AZD2281) were performed by colony formation assay. Data are presented as mean±SEM of three independent experiments. Two-way analysis of variance (ANOVA), **$p < 0.01$. **g** The 293T cells transfected with FLAG-USP15 WT or indicated mutants were lysed and immunoprecipitated by FLAG M2 beads. Samples were analyzed by Western blot with indicated antibodies. **h** USP15-knockout 293T cells were reconstituted with HA-USP15 WT or DM, and treated as indicated for 24 h. Cells were lysed and ubiquitinated BARD1 were pulled down by Ni-NTA under denatured condition and blotted with indicated antibodies. Unprocessed scans of blots are provided in Supplementary Fig. 10. NS not significant

using the keyhole limpet hemocyanin-conjugated peptide. The peptide used for generating phospho-USP15 (Ser678) antibody was: SENEN (pSER)QSEDSVGGC. The antibody was further validated by Western blot and peptide-blocking assay.

**Colony formation assay**. Wild-type or USP15-KO MCF7 cells were seeded at low density and treated with PARP inhibitor (AZD2281). Cells were then cultured at 37 °C for 14 days to allow colonies to form. Colonies were stained with 5% crystal violet/50% methanol. Colonies were counted and statistical data were analyzed by $t$ test analysis. Data were presented as mean ± SEM of three independent experiments.

**Whole-body radiation**. Five-week-old $Usp15^{+/+}$, $Usp15^{+/-}$, and $Usp15^{-/-}$ littermates were randomly allocated and were subjected to 8.5 Gy X-ray radiation. Mice were observed daily for survival and Kaplan–Meier survival curves were generated. In this study, 16 $Usp15^{+/+}$ mice, 25 $Usp15^{+/-}$ mice, and 19 $Usp15^{-/-}$ mice were included.

**Analysis of metaphase chromosomes**. MEF cells were seeded to approximate 70% confluence. Cells were incubated in 20 ng/ml colcemid (Invitrogen) for 2 h. Cells were then harvested and incubated in hypotonic solution (0.06 M KCl) at 37 °C for 30 min. Next, cells were fixed in 10 ml of Carnoy's fixative, and were spread onto pre-cold glass slides and dried. Slides were stained with Giemsa, and 100 metaphase spreads were calculated for each aberration. Data were represented as mean ± SD of three independent experiments.

**Data analysis**. The statistical data were from three biological triplicates. No samples or animals were excluded from the analysis. Statistical analysis was performed by the Student's $t$ test for two groups and by analysis of variance for multiple groups. $P < 0.05$ was considered significant.

**Reporting summary**. Further information on experimental design is available in the Nature Research Reporting Summary linked to this article.

## Data availability

The authors declare that the data support the findings of this study are available within the Supplementary information or from the authors upon reasonable requests. Unprocessed immunofluorescence images have been deposited in the figshare (https://figshare.com/s/66cdec2017fcb3ec877a).

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

## Acknowledgements

We thank Dr. Jeremy Stark at City of Hope, Duarte, CA, USA for U2OS DR-GFP cell line and U2OS EJ5-GFP cell line. We also thank the Imaging Facility of National Center for Protein Sciences-Beijing (NCPSB) for assistance with microscopy. This work was supported by grants from the National Basic Research Program of China (2015CB910601), National Key R&D Program of China (2018YFC0910303), National Natural Science Foundation of China (31371433, 31571463, and 81572740), Beijing Science Program for the Top Young Grant 2015000021223TD04, and the Beijing Municipal Natural Science Foundation (5162010). H.P. was supported by the McCormick Genomic and Proteomic Center of GWU.

## Author contributions

Y.P., Q.L., W.T., C.P., Z.H., Y.C., Z.L., J.L., B.Z., X.L., Y.Y., and Q.S. performed the experiments and analyzed the data. W.Z., C.L., and X.Q. contributed to the critical reagent and in the experimental design. H.P., F.H., and Y.P. designed the study, analyzed the data, and wrote the manuscript.

## Additional information

**Competing interests:** The authors declare no competing interests.

