## [Peer Review File · Nature Communications]

Reviewers' comments:

Reviewer #1 (Remarks to the Author):

This is an interesting manuscript, describing a new molecular mechanism of PARP inhibitor sensitivity caused by loss or mutation of the deubiquitylating enzyme USP15. USP15 deubiquitinates the BARD1 BRCT domain and thereby promotes homologous recombination repair. In contrast, USP15 loss or mutations are associated with increased ubiquitination of DNA repair proteins and subsequent decreased BARD1 BRCT interaction leading to deficiency in homologous recombination and increase sensitivity to PARP inhibition. Mechanistic experiments are carefully conducted, and the interpretation of the results are appropriate. The findings are novel and the paper adds to the current knowledge base of factors influencing homologous recombination and sensitivity to PARP inhibitors.

Reviewer #2 (Remarks to the Author):

The manuscript of Peng et al. provides compelling evidence for the role of USP15 in DNA repair. To their credit the authors have performed a range of experiments wherein the USP15 gene was ablated both in cell lines and in mice, and in both cases the phenotypic effects are convincing. The molecular interactions through which USP15 contributes to homologous recombination repair have been documented through pull downs and mutational studies, and novel findings are reported wherein USP15 phosphorylation by ATM at a specific site is required for DNA repair activity. As discussed in the manuscript these data align with data from cancer repositories and provide potential insights for the dysregulation of USP15 in human cancer. There should be a number of researchers who would find the work of interest. Prior to publication the manuscript requires careful proofreading. It is fraught with typographical errors (for example PARP and BARD1 are spelled incorrectly, the former even in the abstract). Spelling errors extend to figure legends (for example 6a) and should be eliminated. I believe the labels for heterozygous and homozygous null mice have been reversed in Supplementary figure 6c. In the text the terms "knockout" and "knockdown" are used interchangeably, a practice which lacks precision and generates confusion. The methods section is lacking in detail. For example, in what strain was the mouse knockout generated? From the image it would appear that the C57Bl/6 strain was used, but I could not find any text to this effect. It is an important detail given the known variation in knockout phenotypes across strains.

Reviewer #3 (Remarks to the Author):

In this paper, Peng et al. reported the following:

- USP15 promotes homologous recombination.
- USP15 interacts with BARD1 and deubiquitinates the BRCT domain of BARD1.
- Deubiquitination of the BRCT domain of BARD1 facilitates its interaction with HP1 gamma at DSB sites.
- USP15 localizes in the cytoplasm. When DNA damage occurs, USP15 is phosphorylated by ATM, and relocalizes to nuclei. Then, the phosphorylated USP15 interacts with MDC1, accumulates at sites of DNA damage, and deubiquitinates BARD1.
- In some tumor cells, USP15 is mutated, and this mutated USP15 abrogates homologous recombination. This leads to genomic instability and may influence oncogenesis. Tumor cells with decreased USP15 expression are hypersensitive to PARP inhibitors.
- Mutations of the USP15 gene or decreased USP15 expression may be a marker of PARP inhibitor-sensitive cells.

The presented molecular mechanism and potential impact on cancer therapy are of interest. However, there are several concerns regarding the methodology and conclusion.

Major points:

1. Does 53BP1 depletion from USP15 KO cells rescue HR and PARP inhibitor resistance? If USP15 promotes BARD1-BRCA1 retention at DSB sites and promotes DNA end resection, USP15-53BP1 double knockdown (or knockout) cells would show functional HR as well as BRCA1-53BP1 double knockdown cells.
2. <> Because the authors concluded that USP15 regulates the histone H3-HP1gamma-BARD1 axis, they should analyze the efficiency of HR that repairs DNA damage in the genome where the chromatin structure remains intact. Plasmid- based HR and NHEJ assays are not reliable, because multiple copies (may be more than 100) of the reporter plasmid must transfect each cell. Even with the repair of one copy in the cell by HR or NHEJ machinery, the cell would show EGFP fluorescence. The results of the assay are readily influenced by the transfection efficiency. Therefore, the authors should perform the HR assay using cells with just one copy of the reporter gene stably integrated in their genome.
3. <<Cell survival assay>> The authors suggest that USP15 may be a biomarker of cancer treatment using PARP inhibitors. They should confirm the long-term survival potential by colony formation assay. Data solely from an MTS assay are insufficient.
4. <<Immunoblotting in ubiquitination/deubiquitination assay>> Because the authors did not show molecular-weight markers in the immunoblotting panels, readers cannot verify whether the smears actually indicate ubiquitinated BARD1 or ubiquitinated proteins that bind to BARD1.
5. <<KO cell, KD cell, transient expression, stable expression>> Please clearly indicate "gene knockout cells" or "siRNA/shRNA-mediated gene knockdown cells" in each experiment. Please indicate "transient expression" or "stable expression" in each add-back experiment. I could not evaluate the reliability of many experiments.
6. Supplementary Fig. 1a. "p<0.1" is not significant. Therefore, the authors cannot conclude that USP15 KO cells are sensitive to CPT, MMC, or HU.
7. Figure 1. The authors stated: "depletion of USP15 resulted in elevated levels of spontaneous γ H2AX foci formation." However, the difference in the percentage of gamma H2AX-positive cells between control cells and USP15KO cells is very small, and the authors did not perform statistical analysis of this (No IR). How many cells/samples/experiments were analyzed?
8. Supplementary Fig. 1f. Because BARD1 promotes HR by recruiting BRCA1 to sites of DSB, BRCA1 foci should also be analyzed with BARD1 foci. If BRCA1-A complex (BRCA1-Abraxas-RAP80...) foci formation masks USP15-dependent BRCA1 foci formation, the authors can use RAP80 KO or KD cells. I personally think that the data in Supplementary Fig. 1f should be in the main figure, because the results shown in this figure are more important to support the authors' molecular model than the results in Fig. 1e-1h.
9. Figure 2b. I do not see a significant increment of the interaction between BARD1 and Flag-USP15 here. If the authors had loaded a smaller volume of the immunoprecipitated sample, we may have been able to see the difference. The authors may also perform immunoprecipitation of endogenous proteins using anti-USP15 or anti-BARD1 antibody.
10. Figure 2c. Expression levels of USP15-D3 and immunoprecipitated Flag-BARD1 protein in lane D3 are low. This may have affected the results.
11. P9 L9, the authors stated: "On the other hand, we did not observe interaction between USP15 and BRCA1 in vitro (data not shown)." It is very important to confirm whether BRCA1 interacts with USP15. Many readers may be interested in the interaction between BRCA1 and USP15, and so the authors should show the data with appropriate positive (for example, BARD1, Abraxas, or CtIP for endogenous BRCA1 IP) and negative (for example, 53BP1 or RIF1) controls here.
12. Figure 3 and Supplementary Fig. 3 show the expression level of the HA-Ub monomer. Overexpression of the ubiquitin monomer often affects the ubiquitination level of many proteins. The authors should show whole images of the blot stained with HA antibody. Also, show a molecular-weight marker in each panel. This information will enable readers to determine whether the smear actually indicates ubiquitinated BARD1. The authors should try to perform IP with anti-HA antibody and then perform immunoblot with anti-Flag or anti-BARD1 antibody.
13. P9 L18, the authors stated: "More interestingly, we found that USP15, like USP4, can also deubiquitinates itself." The authors did not show sufficient data to support this conclusion. Can

purified USP15 protein deubiquitinate ubiquitinated DUB-dead USP15?

14. P10 L4, the authors stated: "BARD1 ubiquitination slightly decreased after DNA damage" I could not see any difference in the HA-stained smear density between lanes 2 and 3 in Fig. 3a.

15. Figure 3c and 3g. In Fig. 3c, the BRCT ubiquitination level was similar in Control and USP15KO cells. However, in Fig. 3g, BRCT ubiquitination in USP15KO cells was much more marked than that in Control cells. Why did the authors obtain different results?

16. Figure 3g. Based on the authors' model, non-ubiquitinated BRCT of BARD1, but not ubiquitinated BRCT, interacts with HP1gamma. In Fig. 3g, almost the same amount of non-ubiquitinated Flag-BRCT (we can see it as a single band) was immunoprecipitated from each sample and then subjected to the binding assay. Therefore, we should see the same amount of HP1gamma as a binding partner in each sample. However, Flag-BRCT from USP15KO cells interacts with HP1gamma less efficiently than that from control cells. (I am not sure how the authors detected the interaction between them, because the method is not described in the legend. Did they perform IP again after purified Flag-BRCT was incubated with GDT-HP1gamma in vitro? If the authors want to show that ubiquitinated BRCT does not interact with HP1gamma, they can purify ubiquitinated BRCT by tandem IP (anti-Flag and anti-HA) and then compare its binding affinity to HP1gamma before and after in vitro deubiquitination.

17. Fig 3h. The cell lysate used for lane 3 contained more non-ubiquitinated Flag-BRCT than that used for lane 4. Therefore, GST-HP1gamma should pull down more Flag-BRCT from the sample in lane 3 than that in lane 4. However, less non-ubiquitinated Flag-BRCT was detected in lane 3. Can the authors explain why their results deviated from this expectation? They used two kinds of GST proteins (GST-HP1gamma and GST-USP15). This experimental condition is inadequate for the GST pulldown assay. The authors probably detected both USP15-binding BRCT and HP1gamma-binding BRCT.

18. Figure 4g, h. Does USP15 colocalize with MDC1? Is USP15 foci formation abrogated in MDC1 KD or KO cells. If so, would the re-addition of the MDC1 FHA mutant or BRCT mutant to MDC1 KD/KO cells rescue USP15 foci formation or not?

19. Supplementary Fig. 7. The authors may perform BARD1/BRCA1 immunofluorescence to confirm whether BARD1/BRCA1 form foci after IR or PARP inhibitor treatment in cancer cell lines with decreased USP15 expression (MIACAPA2 or PK9) with or without USP15 overexpression. Additionally, they can compare the amount of BARD1/BRCA1 foci among MIACAPA2 (USP15 low), PK9 (USP15 low), and PANC1 (USP15 high). The results generated by these experiments may strongly support the authors' conclusion that the USP15 expression level affects BARD1/BRCA1-dependent HR and PARP inhibitor sensitivity/resistance in cancer cells.

20. P21 L14, the authors stated: "Finally, if small molecule inhibitors of USP15, USP4, and/or USP11 are developed, it will be interesting to pursue their potential in cancer therapy." I think that the advantage of PARP inhibitors is their specificity for cancer cells. If we use USP15 inhibitors clinically, not only cancer cells but also normal cells will be sensitized to PARP inhibitors. This may cause severe side effects. What is the authors' perspective on this?

Minor points:

- P4 "BRAC1" should be "BRCA1."
- P9L1: "(deletion residues 740–952)" should be "(deletion residues 740–981)."
- Supplementary Fig. 1b-e. In the main text, the authors stated: "we examined the accumulation of several DDR factors at DNA lesions induced by UV laser micro-irradiation." However, they showed the results of IR-ed samples.
- Supplementary Fig. 2b. (P8 L16). Here, the authors should perform GST pull down using full-length GST-BARD1.
- Supplementary Fig. 7a: "Deep delition" may be "Deep deletion."
- Methods: Indicate the RNF8 and RNF168 antibodies used in the study. The authors should also indicate antibody dilutions for WB and IF.
- P11 L10, the authors stated: "we knocked down USP15", but USP15 knockout cells were used in Fig. 3e-g.
- P12 L6, "double knockdown": In Fig. 4f, g, the authors used USP15 knockout cells and BARD1

siRNA.

- P16 L21: "Fig. 7f" should be "Fig. 7g."
- P17 L1: "Fig. 7g" should be "Fi. 7f."
- P29: "CtiP" should be "CtIP."

Reviewer #4 (Remarks to the Author):

Peng et al. provide compelling evidence for the deubiquitinase USP15 as a novel modulator of homologous recombination and genome stability. A thorough mechanistic dissection uncovers ATM-mediated USP15 phosphorylation as a mediator of USP15 interaction with MDC1 and subsequent association with and deubiquitination of BARD1. The latter affects HP1g interaction with BARD1. Underlining functional relevance, USP15 deficiency causes defects in end resection and HR as well as increased PARPi sensitivity. However, it remains to be demonstrated if impaired HP1 recruitment is sufficient to account for the observed HR defect. A knockout mouse model corroborates the impact of USP15 on genome stability and TCGA analyses suggest a broader role for USP15 mutations in tumor formation. Although correlative at this point, it is intriguing that tumor-relevant mutations occur in the ATM targeted serine and the BARD1 interacting domain, suggesting that they may act at least in part by increasing genome instability. Together, this study establishes USP15 as an important and cancer-relevant mediator of genome maintenance via HR.

The following issues should be addressed prior to acceptance:

- 1) The authors show impaired recruitment of HR effectors to sites of laser-induced damage (Fig 1 e-h). In contrast accumulation of upstream effectors such as RNF8, FK2 and MDC1 appears unaffected (Fig S1 b-e). However, the latter is based solely on IR. To allow for a direct comparison of recruitment defects, the authors will need to repeat the analyses in Figure S1 in response to laser-induced DNA damage formation.
- 2) The nature of the poly-ubiquitination targeted by USP15 is unclear and needs to be further investigated. Is BARD1 stabilized upon USP15 over-expression and/or phosphorylation? Are the ubiquitin chains K48- or K63-linked? Does proteasome inhibition have any impact on the effects observed upon USP21 loss? Is the adverse effect of BARD1 ubiquitination on HP1 recruitment direct or an indirect consequence of decreased BARD1 protein?
- 3) Given the proposed working model, it would be important to show that HP1 is epistatic with USP15.
- 4) USP15 accumulation at sites of IR-induced damage is hard to see (Fig. 4e). The authors should show intensity sections across gH2AX foci to corroborate this point. Similarly, IR-induced foci, on which many of the quantifications are based, are hard to discern. Example images for foci quantification should be included, and some experiments should be repeated at a lower IR dose (< 5 Gy) to reduce the damage load.
- 5) Page numbers are missing, but the statement that "in Fig 1a, depletion of USP15 resulted in elevated levels of spontaneous gH2AX foci formation" is not supported by the data.
- 6) Cell cycle analyses should be included in the supplemental material. Based on the mouse phenotype, a proliferative defect may be expected. Similarly, the effect on 53BP1 needs to be shown.
- 7) Several of the figures are mis-referenced in the text, e.g. Fig 7g should be 7f, and failure to deubiquitinate BARD1 is not shown in this panel.
- 8) DUSP needs to be defined, and more generally, the introduction of USP15 is rather short.
- 9) Nuclear intensities in Fig. S5b need to be quantified.
- 10) The weight change in the KO mice is not shown in Fig. S6d.
- 11) The foci data in MEFs are not convincing.
- 12) The authors' speculation on why BARD1 ubiquitination did not increase in USP15 knockdown cells in the absence of damage is not very satisfying (page ???). ATM dependence is a more likely explanation.

Response to referees:

We thank reviewers for their constructive comments and positive evaluations on our study. In the revised version, we have carefully considered the reviewers' criticisms and further improved our manuscript.

Reviewer #1 (Remarks to the Author):

This is an interesting manuscript, describing a new molecular mechanism of PARP inhibitor sensitivity caused by loss or mutation of the deubiquitylating enzyme USP15. USP15 deubiquitinates the BARD1 BRCT domain and thereby promotes homologous recombination repair. In contrast, USP15 loss or mutations are associated with increased ubiquitination of DNA repair proteins and subsequent decreased BARD1

BRCT interaction leading to deficiency in homologous recombination and increase sensitivity to PARP inhibition. Mechanistic experiments are carefully conducted, and the interpretation of the results are appropriate. The findings are novel and the paper adds to the current knowledge base of factors influencing homologous recombination and sensitivity to PARP inhibitors.

Response: We thank the reviewer for his/her positive comments on our manuscript. We also hope our manuscript will bring new insights toward homologous recombination and cancer chemotherapy.

Reviewer #2 (Remarks to the Author):

The manuscript of Peng et al. provides compelling evidence for the role of USP15 in DNA repair. To their credit the authors have performed a range of experiments wherein the USP15 gene was ablated both in cell lines and in mice, and in both cases the phenotypic effects are convincing. The molecular interactions through which USP15 contributes to homologous recombination repair have been documented through pull downs and mutational studies, and novel findings are reported wherein USP15 phosphorylation by ATM at a specific site is required for DNA repair activity. As discussed in the manuscript these data align with data from cancer repositories and provide potential insights for the dysregulation of USP15 in human cancer. There should be a number of researchers who would find the work of interest. Prior to publication the manuscript requires careful proofreading. It is fraught with typographical errors (for example PARP and BARD1 are spelled incorrectly, the former even in the abstract). Spelling errors extend to figure legends (for example 6a) and should be eliminated. I believe the labels for heterozygous and homozygous null mice have been reversed in Supplementary figure 6c. In the text the terms "knockout" and "knockdown" are used interchangeably, a practice which lacks precision and generates confusion. The methods section is lacking in detail. For example, in what strain was the mouse knockout generated? From the image it would appear that the C57Bl/6 strain was used, but I could not find any text to this effect. It is an important detail given the known variation in knockout phenotypes across strains.

Response: We thank the reviewer for positive comments on our manuscript. We have carefully revised the manuscript and corrected typographical errors. We used the

C57BL/6J strain to generate *Usp15* knockout mice, and we have included this information in the method section of the revised version.

Reviewer #3 (Remarks to the Author):

In this paper, Peng et al. reported the following: USP15 promotes homologous recombination. USP15 interacts with BARD1 and deubiquitinates the BRCT domain of BARD1. Deubiquitination of the BRCT domain of BARD1 facilitates its interaction with HP1 gamma at DSB sites. USP15 localizes in the cytoplasm. When DNA damage occurs, USP15 is phosphorylated by ATM, and relocates to nuclei. Then, the phosphorylated USP15 interacts with MDC1, accumulates at sites of DNA damage, and deubiquitinates BARD1. In some tumor cells, USP15 is mutated, and this mutated USP15 abrogates homologous recombination. This leads to genomic instability and may influence oncogenesis. Tumor cells with decreased USP15 expression are hypersensitive to PARP inhibitors. Mutations of the USP15 gene or decreased USP15 expression may be a marker of PARP inhibitor sensitive cells. The presented molecular mechanism and potential impact on cancer therapy are of interest. However, there are several concerns regarding the methodology and conclusion.

Response: We really appreciate the professional comment and a positive evaluation on our study. In the revised version, in the light of the constructive comments of the referee, we further improve our manuscript.

Major points:

1. Does 53BP1 depletion from USP15 KO cells rescue HR and PARP inhibitor resistance? If USP15 promotes BARD1BRCA1 retention at DSB sites and promotes DNA end resection, USP15 53BP1 double knockdown (or knockout) cells would show functional HR as well as BRCA153BP1 double knockdown cells.

Response: We completely agree with the reviewer's comment. As shown in Supplementary Fig. 1i and 1j, 53BP1 depletion from USP15 KO cells rescued HR and PARP inhibitor resistance. These results further support that USP15 promotes DNA end resection and HR repair.

2. <> *Because the authors concluded that USP15 regulates the histone H3HP1gammaBARD1 axis, they should analyze the efficiency of HR that repairs DNA damage in the genome where the chromatin structure remains intact. Plasmid based HR and NHEJ assays are not reliable, because multiple copies (may be more than 100) of the reporter plasmid must transfect each cell. Even with the repair of one copy in the cell by HR or NHEJ machinery, the cell would show EGFP fluorescence. The results of the assay are readily influenced by the transfection efficiency. Therefore, the authors should perform the HR assay using cells with just one copy of the reporter gene stably integrated in their genome.*

Response: In the light of the constructive suggestion of the referee, we got U2OS-DR-GFP and U2OS-EJ5-GFP cells from Dr. Jeremy Stark's lab. We have repeated all the HR and NHEJ assays using these integration reporter systems. As shown in new Fig. 1b-c, Fig. 2e, Fig. 5b, Fig. 7b, Supplementary Fig. 1i, Supplementary Fig. 3a, and Supplementary Fig. 4h, USP15 regulated HR, and had no significant effect on NHEJ.

3. <<Cell survival assay>> *The authors suggest that USP15 may be a biomarker of cancer treatment using PARP inhibitors. They should confirm the long term survival potential by colony formation assay. Data solely from an MTS assay are insufficient.*

Response: We have performed colony formation assay. As shown in new Fig. 2f, Fig. 5a, Fig. 7f, Supplementary Fig. 3b and Supplementary Fig. 7d-f, USP15 could be a biomarker of cancer treatment using PARP inhibitors.

4. <<Immunoblotting in ubiquitination/deubiquitination assay>> *Because the authors did not show molecular weight markers in the immunoblotting panels, readers cannot verify whether the smears actually indicate ubiquitinated BARD1 or ubiquitinated proteins that bind to BARD1.*

Response: As suggested by the reviewer, we have added molecular weight markers in the immunoblotting panels.

5. <<KO cell, KD cell, transient expression, stable expression>> *Please clearly indicate "gene knockout cells" or "siRNA/shRNA mediated gene knockdown cells" in*

each experiment. Please indicate “transient expression” or “stable expression” in each addback experiment. I could not evaluate the reliability of many experiments.

Response: We have now indicated the detailed information of each experiment clearly in figure legend and method sections.

6. Supplementary Fig.1a. “ $p < 0.1$ ” is not significant. Therefore, the authors cannot conclude that USP15 KO cells are sensitive to CPT, MMC, or HU.

Response: We are sorry for our carelessness. “*” should be indicated as “ $p < 0.01$ ”, and we have corrected this error in the revised version.

7. Figure 1. The authors stated: “depletion of USP15 resulted in elevated levels of spontaneous γ H2AX foci formation.” However, the difference in the percentage of gamma H2AXpositive cells between control cells and USP15KO cells is very small, and the authors did not perform statistical analysis of this (No IR). How many cells/samples/experiments were analyzed?

Response: To do the quantification, we have conducted 3 independent experiments. For each experiment, we analyzed 100 cells. We did the statistical analysis. As shown in Letter Fig. 1, depletion of USP15 resulted in elevated levels of spontaneous γ H2AX foci formation ($p < 0.05$).

Letter Fig.1. Analysis of γ H2AX foci formation without IR treatment. $p < 0.05$.

8. Supplementary Fig. 1f. Because BARD1 promotes HR by recruiting BRCA1 to sites of DSB, BRCA1 foci should also be analyzed with BARD1 foci. If BRCA1A complex (BRCA1AbraxasRAP80...) foci formation masks USP15dependent BRCA1 foci formation, the authors can use RAP80 KO or KD cells. I personally think that the data in Supplementary Fig.1f should be in the main figure, because the results shown

in this figure are more important to support the authors' molecular model than the results in Fig. 1e1h.

Response: We thank the reviewer for the insightful suggestion. We have moved these data to the new Fig. 1i. USP15 deficient cells also exhibited impaired BRCA1 foci formation, but had no effect on RAP80 recruitment (Supplementary Fig. 1g).

9. *Figure 2b. I do not see a significant increment of the Interaction between BARD1 and FlagUSP15 here. If the authors had loaded a smaller volume of the immunoprecipitated sample, we may have been able to see the difference. The authors may also perform immunoprecipitation of endogenous proteins using antiUSP15 or antiBARD1 antibody.*

Response: We have repeated this experiment as the reviewer suggested. We performed immunoprecipitation of endogenous proteins using anti-BARD1 antibody. As shown in the new Fig.2b, the interaction between BARD1 and USP15 was induced upon DNA damage.

10. *Figure 2c. Expression levels of USP15 D3 and immunoprecipitated FlagBARD1 protein in lane D3 are low. This may have affected the results.*

Response: We have repeated this assay. As shown in new Fig. 2c, USP15 D3 mutant still did not interact with BARD1.

11. *P9 L9, the authors stated: "On the other hand, we did not observe interaction between USP15 and BRCA1 in vitro (data not shown)." It is very important to confirm whether BRCA1 interacts with USP15. Many readers may be interested in the interaction between BRCA1 and USP15, and so the authors should show the data with appropriate positive (for example, BARD1, Abraxas, or CtIP for endogenous BRCA1 IP) and negative (for example, 53BP1 or RIF1) controls here.*

Response: This is really a good suggestion, and we have performed endogenous BRCA1 immunoprecipitation. As shown in new Supplementary Fig. 2h, BRCA1 did not bind USP15 significantly.

12. *Figure 3 and Supplementary Fig. 3 show the expression level of the HAUb monomer. Overexpression of the ubiquitin monomer often affects the ubiquitination*

level of many proteins. The authors should show whole images of the blot stained with HA antibody. Also, show a molecular weight marker in each panel. This information will enable readers to determine whether the smear actually indicates ubiquitinated BARD1. The authors should try to perform IP with antiHA antibody and then perform immunoblot with antiFlag or antiBARD1 antibody.

Response: We thank the reviewer for the professional suggestion. In our new Fig.3a, c, d, Supplementary Fig. 5f, h and Fig.7h, we overexpressed His tagged ubiquitin in cells, performed pull down of ubiquitinated proteins with Ni-NTA and then performed immunoblot with anti-FLAG antibody. The detailed methods were described in the method section. We also included the whole images of the blot panels indicating the ubiquitination levels, as well as labeling the molecular weight marker.

13. P9 L18, the authors stated: "More interestingly, we found that USP15, like USP4, can also deubiquitinates itself." The authors did not show sufficient data to support this conclusion. Can purified USP15 protein deubiquitinate ubiquitinated DUB dead USP15?

Response: We overexpressed and purified ubiquitinated DUB dead USP15 from 293T cells, and performed *in vitro* deubiquitination assay. As shown in new Supplementary Fig. 3h-i, purified wild type USP15 protein can deubiquitinate ubiquitinated DUB dead USP15, while purified DUB dead USP15 can not.

14. P10 L4, the authors stated: "BARD1 ubiquitination slightly decreased after DNA damage" I could not see any difference in the HA stained smear density between lanes 2 and 3 in Fig. 3a.

Response: We have repeated this experiment using His Ub pull down assay. As shown in our new Fig. 3a, BARD1 ubiquitination decreased slightly after DNA damage.

15. Figure 3c and 3g. In Fig. 3c, the BRCT ubiquitination level was similar in Control and USP15KO cells. However, in Fig. 3g, BRCT ubiquitination in USP15KO cells was much more marked than that in Control cells. Why did the authors obtain different results?

Response: We treated cells with MMC in old Fig. 3g, and USP15 depletion prevented BRCT ubiquitination reduction following DNA damage. In old Fig. 3c, the BRCT ubiquitination level was similar in Control and USP15KO cells without MMC treatment, and USP15 depletion still prevented BRCT ubiquitination reduction after DNA damage (compare lane 3 and lane5 in new Fig. 3c). We labeled the figures more clearly in the revised version.

16. *Figure 3g. Based on the authors' model, nonubiquitinated BRCT of BARD1, but not ubiquitinated BRCT, interacts with HP1gamma. In Fig. 3g, almost the same amount of nonubiquitinated FlagBRCT (we can see it as a single band) was immunoprecipitated from each sample and then subjected to the binding assay. Therefore, we should see the same amount of HP1gamma as a binding partner in each sample. However, FlagBRCT from USP15KO cells interacts with HP1gamma less efficiently than that from control cells. (I am not sure how the authors detected the interaction between them, because the method is not described in the legend. Did they perform IP again after purified FlagBRCT was incubated with GSTHP1gamma in vitro? If the authors want to show that ubiquitinated BRCT does not interact with HP1gamma, they can purify ubiquitinated BRCT by tandem IP (antiFlag and antiHA) and then compare its binding affinity to HP1gamma before and after in vitro deubiquitination.*

17. *Fig 3h. The cell lysate used for lane 3 contained more nonubiquitinated FlagBRCT than that used for lane 4. Therefore, GSTHP1gamma should pull down more FlagBRCT from the sample in lane 3 than that in lane 4. However, less nonubiquitinated FlagBRCT was detected in lane 3. Can the authors explain why their results deviated from this expectation? They used two kinds of GST proteins (GSTHP1gamma and GSTUSP15). This experimental condition is inadequate for the GST pulldown assay. The authors probably detected both USP15binding BRCT and HP1gammabinding BRCT.*

Response to question 16 and 17: We thank the reviewer for these insightful comments. Indeed, in old Fig. 3g and 3h, we want to show that non-ubiquitinated BRCT of BARD1, but not ubiquitinated BRCT, interacts with HP1gamma *in vitro*. In the light of the good suggestion of the referee, we redid this *in vitro* binding assay. Briefly, ubiquitinated BARD1-BRCT were purified from 293T cells by tandem IP (first HA

beads and then FLAG beads) and were left on the FLAG beads. Then one half part was left untreated and one half part was deubiquitinated by His-USP15 *in vitro*. Next, the immunoprecipitates were washed with PBS three times and were then incubated with GST or GST-HP1 γ *in vitro*. As shown in new Fig. 3g, USP15 mediated deubiquitination of BARD1 BRCT domain promoted BARD1-HP1 γ interaction. This result fits with our hypothesis.

18. *Figure 4g, h. Does USP15 colocalize with MDC1? Is USP15 foci formation abrogated in MDC1 KD or KO cells. If so, would the readdition of the MDC1 FHA mutant or BRCT mutant to MDC1 KD/KO cells rescue USP15 foci formation or not?*

Response: We thank the reviewer for these constructive suggestions. In order to better address the reviewer's concerns, we have generated a phospho-specific antibody against USP15 S678. We validated the antibodies using USP15 S678A mutant and specific peptide competition assays (Supplementary Fig. 5d-e). Upon DNA damage, phosphorylated USP15 was recruited to DNA damage sites to form foci, which co-localized with both MDC1 and γ H2AX (Fig. 4f, g). Furthermore, MDC1 depletion abrogated USP15 foci formation (Fig. 4g), and re-introduction of the WT MDC1 but not FHA domain deletion mutant could rescue USP15 foci formation (Fig. 4k). Of note, BRCT domain deletion mutant of MDC1 could not be recruited to DNA damage sites, USP15 foci formation was also impaired in these cells (Fig. 4k).

19. *Supplementary Fig. 7. The authors may perform BARD1/BRCA1 immunofluorescence to confirm whether BARD1/BRCA1 form foci after IR or PARP inhibitor treatment in cancer cell lines with decreased USP15 expression (MIACAPA2 or PK9) with or without USP15 overexpression. Additionally, they can compare the amount of BARD1/BRCA1 foci among MIACAPA2 (USP15 low), PK9 (USP15 low), and PANC1 (USP15 high). The results generated by these experiments may strongly support the authors' conclusion that the USP15 expression level affects BARD1/BRCA1 dependent HR and PARP inhibitor sensitivity/resistance in cancer cells.*

Response: We thank the reviewer for this interesting suggestion. As shown in Supplementary Fig. 7g-i, we investigated the BARD1/BRCA1 foci formation after PARP inhibitor treatment in MIAPACA2 and PK9 cells with or without USP15

overexpression. Overexpression of USP15 in these two pancreatic cancer cells increased BARD1/BRCA1 foci formation after PARP inhibitor treatment (Supplementary Fig. 7g, h), and depletion of USP15 in PANC1 cells impaired BARD1/BRCA1 foci formation (Supplementary Fig. 7i). Additionally, MIAPACA2 and PK9 cells (USP15 low) showed less BARD1/BRCA1 foci than PANC1 cells (USP15 high). These results further support that the USP15 expression level affects BARD1/BRCA1-dependent HR and PARP inhibitor sensitivity/resistance in cancer cells.

20. P2 1 L14, the authors stated: “Finally, if small molecule inhibitors of USP15, USP4, and/or USP11 are developed, it will be interesting to pursue their potential in cancer therapy.” I think that the advantage of PARP inhibitors is their specificity for cancer cells. If we use USP15 inhibitors clinically, not only cancer cells but also normal cells will be sensitized to PARP inhibitors. This may cause severe side effects. What is the authors’ perspective on this?

Response: We completely agree with the reviewer’s perspective. We deleted this sentence in the revised version.

Minor points:

P4 “BRAC1” should be “BRCA1.”

P9L1: “(deletion residues 740–952)” should be “(deletion residues 740–981).”

Supplementary Fig. 1be. In the main text, the authors stated: “we examined the accumulation of several DDR factors at DNA lesions induced by UV laser microirradiation.” However, they showed the results of IRed samples.

Supplementary Fig. 2b. (P8 L16). Here, the authors should perform GST pull down using full length GSTBARD1.

Supplementary Fig. 7a: “Deep delition” may be “Deep deletion.”

Methods: Indicate the RNF8 and RNF168 antibodies used in the study. The authors should also indicate antibody dilutions for WB and IF.

P11 L10, the authors stated: “we knocked down USP15”, but USP15 knockout cells were used in Fig. 3eg.

P12 L6, “double knockdown”: In Fig. 4f, g, the authors used USP15 knockout cells and BARD1 siRNA.

P16 L21: “Fig. 7f” should be “Fig. 7g.”

P17 L1: “Fig. 7g” should be “Fi. 7f.”

Response to minor points: We are sorry for our carelessness, and we have fixed all the problems in the revised version:

- 1, We have proofread our manuscript carefully, and corrected typographic errors.
- 2, We have repeated the analyses in Figure S1 in response to laser induced DNA damage formation. As shown in new Supplementary Fig. 1d-g, the accumulation of upstream effectors of BRCA1, such as RNF8, FK2, MDC1, and RAP80, were not affected in USP15 knockout cells.
- 3, We performed GST pull down assay using the full length of GST-BARD1 as the reviewer suggested, and we have included this data in the new Supplementary Fig. 2b.
- 4, We provided the detailed information about antibody dilutions for WB and IF in the method section.

Reviewer #4 (Remarks to the Author):

Peng et al. provide compelling evidence for the deubiquitinase USP15 as a novel modulator of homologous recombination and genome stability. A thorough mechanistic dissection uncovers ATM mediated USP15 phosphorylation as a mediator of USP15 interaction with MDC1 and subsequent association with and deubiquitination of BARD1. The latter affects HP1g interaction with BARD1. Underlining functional relevance, USP15 deficiency causes defects in end resection and HR as well as increased PARPi sensitivity. However, it remains to be demonstrated if impaired HP1 recruitment is sufficient to account for the observed HR defect. A knockout mouse model corroborates the impact of USP15 on genome stability and TCGA analyses suggest a broader role for USP15 mutations in tumor formation. Although correlative at this point, it is intriguing that tumor relevant mutations occur in the ATM targeted serine and the BARD1 interacting domain, suggesting that they may act at least in part by increasing genome instability. Together, this study establishes USP15 as an important and cancer relevant mediator of genome maintenance via HR.

Response: We thank the reviewer for the positive comments on our manuscript. We have addressed the reviewer’s concerns as described below.

The following issues should be addressed prior to acceptance:

1) The authors show impaired recruitment of HR effectors to sites of laser induced damage (Fig 1 eh). In contrast accumulation of upstream effectors such as RNF8, FK2 and MDC1 appears unaffected (Fig S1 be). However, the latter is based solely on IR. To allow for a direct comparison of recruitment defects, the authors will need to repeat the analyses in Figure S1 in response to laser induced DNA damage formation.

Response: We have repeated the analyses in Figure S1 in response to laser induced DNA damage formation. As shown in new Supplementary Fig. 1d-g, the accumulation of upstream effectors of BRCA1, such as RNF8, FK2, MDC1, and RAP80, were not affected in USP15 knockout cells.

2) The nature of the polyubiquitination targeted by USP15 is unclear and needs to be further investigated. Is BARD1 stabilized upon USP15 overexpression and/or phosphorylation? Are the ubiquitin chains K48 or K63 linked? Does proteasome inhibition have any impact on the effects observed upon USP21 loss? Is the adverse effect of BARD1 ubiquitination on HP1 recruitment direct or an indirect consequence of decreased BARD1 protein?

Response: This is really a good question. We found that BARD1 was not stabilized upon USP15 overexpression (Letter Fig. 2a). We reintroduced WT USP15, S678A mutant or S678D mutant into USP15 depleted 293T cells, and treated cells with or without IR. As shown in Letter Fig. 2b, BARD1 expression level did not change upon USP15 S678 phosphorylation. Moreover, we treated 293T cells with ATMi (Ku55933) with or without IR (Letter Fig. 2c), USP15 phosphorylation did not affect BARD1 level again. We also confirmed that USP15 regulated the K63 linked ubiquitin chain of BARD1 BRCT domain, but not K48 linked ubiquitin chain (Fig. 3b). K48-linked Ub chains are associated with protein degradation, while K63-linked chains are mainly non-degradative and have signaling roles. The above results strongly suggest that USP15 affects BARD1 function but not proteasome dependent protein turnover.

Letter. Fig. 2. The stability of BARD1 was not regulated by USP15.

3) Given the proposed working model, it would be important to show that HP1 is epistatic with USP15.

Response: As shown in our new Supplementary Fig. 4h-i, double depletion of USP15 and HP1 γ showed similar HR efficiency/PARP inhibitor sensitivity as compared to either single depletion. These results suggested that HP1 γ is epistatic with USP15.

4) USP15 accumulation at sites of IR induced damage is hard to see (Fig. 4e). The authors should show intensity sections across γ H2AX foci to corroborate this point. Similarly, IR induced foci, on which many of the quantifications are based, are hard to discern. Example images for foci quantification should be included, and some experiments should be repeated at a lower IR dose (< 5 Gy) to reduce the damage load.

Response: We have generated and a phospho-specific antibody against S678. We validated the antibodies using USP15 S678A mutant and specific peptide competition assays (Supplementary Fig. 5d-e). Upon DNA damage, phosphorylated USP15 was recruited to DNA damage sites to form clear foci, which co-localized with both MDC1 and γ H2AX very well (Fig. 4f, g). Furthermore, MDC1 depletion abrogated USP15 foci formation (Fig. 4g), and re-introduction of the WT MDC1 but not FHA domain deletion mutant could rescue USP15 foci formation (Fig. 4k). We repeated the foci formation experiments at a lower IR dose (4Gy), and got the similar results. We also included the example images for foci quantification in the revised version (Fig. 1i, Fig. 5c-f, Fig. 7c-e, Supplementary Fig. 2c-f, Supplementary Fig. 3c-f, Supplementary Fig. 6e-g).

5) Page numbers are missing, but the statement that “in Fig 1a, depletion of USP15 resulted in elevated levels of spontaneous gH2AX foci formation” is not supported by the data.

Response: We have conducted the statistical analysis of this data. And statistical analysis supported that depletion of USP15 resulted in elevated levels of spontaneous gH2AX foci formation.

6) Cell cycle analyses should be included in the supplemental material. Based on the mouse phenotype, a proliferative defect may be expected. Similarly, the effect on 53BP1 needs to be shown.

Response: We have included the cell cycle analyses and 53BP1 foci formation data in the new supplementary Fig. 1c and Supplementary Fig. 1h respectively.

7) Several of the figures are misreferenced in the text, e.g. Fig 7g should be 7f, and failure to deubiquitinate BARD1 is not shown in this panel.

Response: We are sorry for our carelessness, and we have corrected these errors.

8) DUSP needs to be defined, and more generally, the introduction of USP15 is rather short.

Response: We have provided more background about DUSP and USP15 in the main text.

9) Nuclear intensities in Fig. S5b need to be quantified.

Response: We provided this quantification data in the new supplementary Fig. 5c. The quantification data further supported our conclusion.

10) The weight change in the KO mice is not shown in Fig. S6d.

Response: We have included this data in the Supplementary Fig. 6d.

11) The foci data in MEFs are not convincing.

Response: We have repeated these experiments, provided representative images and performed quantification analysis. As shown in Supplementary Fig. 6e-h. IR induced BARD1, RPA, and RAD51 foci formation were impaired in *Usp15*^{-/-} MEF cells.

12) The authors' speculation on why BARD1 ubiquitination did not increase in USP15 knockdown cells in the absence of damage is not very satisfying (page ???). ATM dependence is a more likely explanation.

Response: This is really a thoughtful suggestion. Indeed, ATM inhibitor (Ku55933) treatment inhibited USP15 mediated BARD1 deubiquitination upon DNA damage (Supplementary Fig. 5h). Furthermore, USP15 S678A mutant didn't deubiquitinate BARD1-BRCT upon DNA damage (Supplementary Fig. 5f). These data supported that USP15 mediated BARD1 deubiquitination was ATM dependent.

REVIEWERS' COMMENTS:

Reviewer #3 (Remarks to the Author):

The authors have satisfactorily addressed my comments.

Reviewer #4 (Remarks to the Author):

The authors have addressed my concerns. It would, however, be helpful to discuss the findings in the context of other recent work implicating USPs in HR (e.g. PMID: 29335415, PMID: 28743957, PMID: 28495793 and others).

REVIEWERS' COMMENTS:

Reviewer #3 (Remarks to the Author):

The authors have satisfactorily addressed my comments.

Response: Thank you!

Reviewer #4 (Remarks to the Author):

The authors have addressed my concerns. It would, however, be helpful to discuss the findings in the context of other recent work implicating USPs in HR (e.g. PMID: 29335415, PMID: 28743957, PMID: 28495793 and others).

Response: Thank you! We have included more discussions in the main text.